# TARGET-AWARE VIDEO DIFFUSION MODELS

**Taeksoo Kim**
Seoul National University
taeksu98@snu.ac.kr

**Hanbyul Joo**[†]
Seoul National University & RLWRLD
hbjoo@snu.ac.kr

## ABSTRACT

We present a target-aware video diffusion model that generates videos from an input image, in which an actor interacts with a specified target while performing a desired action. The target is defined by a segmentation mask, and the action is described through a text prompt. Our key motivation is to incorporate target awareness into video generation, enabling actors to perform directed actions on designated objects. This enables video diffusion models to act as motion planners, producing plausible predictions of human-object interactions by leveraging the priors of large-scale video generative models. We build our target-aware model by extending a baseline model to incorporate the target mask as an additional input. To enforce target awareness, we introduce a special token that encodes the target's spatial information within the text prompt. We then fine-tune the model with our curated dataset using an additional cross-attention loss that aligns the cross-attention maps associated with this token with the input target mask. To further improve performance, we selectively apply this loss to the most semantically relevant attention regions and transformer blocks. Experimental results show that our target-aware model outperforms existing solutions in generating videos where actors interact accurately with the specified targets. We further demonstrate its efficacy in two downstream applications: zero-shot 3D HOI motion synthesis with physical plausibility and long-term video content creation.

## 1 INTRODUCTION

Video diffusion models have demonstrated remarkable capabilities in simulating complex real-world scenes. Ideally, such models can serve as motion planners, in line with the concept of world models (Ha & Schmidhuber, 2018; Bar et al., 2024; Kim et al., 2025a), by producing plausible predictions of interactions between an actor (human or robot) and target objects using priors learned from large-scale video datasets. However, existing image-to-video diffusion models (Yang et al., 2025b; Kong et al., 2024; HaCohen et al., 2024), which generate videos from an input image guided by text prompts, are target-unaware. An alternative line of work attempts to explicitly control actor-target interactions using dense structural or motion cues, such as depth maps (Esser et al., 2023; Zhang et al., 2024), edges (Chen et al., 2023; Khachatryan et al., 2023), optical flow (Ni et al., 2023; Burgert et al., 2025; Gu et al., 2025), motion trajectories (Yan et al., 2023; Shi et al., 2024a), or drag-based manipulation (Deng et al., 2024; Teng et al., 2023; Shi et al., 2024b). While effective for certain tasks, these approaches do not meet our needs. Our goal is to use video diffusion models to infer plausible actor-target interactions, where action guidance for the actor is not available in advance and thus cannot be provided as input. Ultimately, we aim to leverage video generative models for high-level action planning, inferring realistic interaction cues for the actor within the current scene, as explored in recent robotics research (Black et al., 2024; Du et al., 2023; Ajay et al., 2023; Ni et al., 2024).

In this paper, we present a target-aware video diffusion model that generates videos from an input image, where an actor performs a desired action directed at a specified target. The target is defined by a segmentation mask, and the action is described with a text prompt. We use the mask as a means to specify the target object in the scene, which can be obtained with minimal effort (e.g., a single click),

---

[†]Corresponding author

Project page: https://taeksuu.github.io/tavid/

or automatically from text input using off-the-shelf tools (Ren et al., 2024), and we show that our method is robust to variations in mask quality. By providing an explicit way to designate the target, our model can serve as an effective motion planner, enabling the actor to perform diverse interactions with the specified object. While our training uses videos with human actors, we also demonstrate that the model generalizes seamlessly to other agents, including animals and robotic hands.

To integrate spatial information of the target mask, we extend a base image-to-video diffusion model (Yang et al., 2025b) to take the mask as an additional input, and fine-tune it on our newly curated dataset. However, simply fine-tuning the model with the extra mask input does not ensure the target awareness of the model. To address this, we introduce a special token, [TGT], into the text prompt to describe the target and enforce an alignment between the [TGT] token's cross-attention maps and the input target mask by applying a loss on the model's cross-attention. This cross-attention loss enables the model to associate the [TGT] token with the spatial information of the target, improving the precision of the generated interactions with the target by injecting spatial grounding into the text-conditioning mechanism of the model. We selectively apply this loss to specific attention regions and transformer blocks that are most semantically relevant for effective supervision.

Experimental results demonstrate that our target-aware video diffusion model outperforms existing solutions in synthesizing videos where actors precisely engage with designated targets. To further demonstrate the strength of our target-aware model, we apply our model to two downstream applications: (1) zero-shot 3D human-object interaction (HOI) motion synthesis with physical plausibility, simulating physical agents performing plausible actions in a given environment, and (2) video content creation, generating long-term videos covering navigations and interactions with minimal user input.

Our contributions can be summarized as follows: (1) We present a target-aware video diffusion model that generates videos of interactions between the actor and the target using a segmentation mask and a text prompt; (2) We propose to utilize a cross-attention loss to enable the base model to effectively incorporate the mask input and achieve target awareness, and provide a comprehensive analysis of its effects across different parts of the model; (3) We present a newly curated dataset specifically designed to train and evaluate our target-aware model; and (4) We demonstrate two real-world applications of our target-aware model: zero-shot 3D HOI motion synthesis for controlling physical agents and video content creation.

## 2 RELATED WORK

**Controllable Video Generation** Building on early work (Ho et al., 2022; Esser et al., 2023; Guo et al., 2024) that extends text-to-image diffusion models to video generation, the community has shown significant interest in producing videos with enhanced controls. Several methods, inspired by ControlNet (Zhang et al., 2023b), have been adapted for videos, where structural cues, such as depth maps (Esser et al., 2023; Zhang et al., 2024), edge information (Chen et al., 2023; Khachatryan et al., 2023), optical flow (Ni et al., 2023; Gu et al., 2025), or motion (Yan et al., 2023; Shi et al., 2024a; Cha et al., 2026), are integrated into the generation process via additional modules to produce structure-consistent outputs. Another line of research focuses on manipulating the internal representations of diffusion models to achieve controls without extra modules. Attention modulation approaches (Yang et al., 2024; Wu et al., 2024a; Jain et al., 2024) adjust cross-attention maps of predefined regions to steer subject movements. Inversion-based feature injection methods (Liu et al., 2024; Wang et al., 2023a; Jeong & Ye, 2024) edit videos in a zero-shot manner by leveraging cross-attention control, allowing for content editing without training. Other work (Teng et al., 2023; Deng et al., 2024; Wu et al., 2024b; Yin et al., 2023) extends drag-based image editing techniques (Pan et al., 2023; Shi et al., 2024b; Shin et al., 2024) to video, by either adding an extra drag-embedding module or optimizing video latents to align with drag inputs. While existing approaches focus on generating videos that faithfully follow the dense input cues, either from source video or user inputs, we aim to extract those motion cues from video diffusion models with minimal extra input, a mask of the target.

**Human-Scene Interaction Synthesis.** Synthesizing natural human motions within a given scene remains challenging, requiring a high-level semantic understanding of human-scene interactions and affordances. Early work primarily focuses on posing a static 3D human in a 3D environment (Kim et al., 2014; Savva et al., 2016; Li et al., 2019; Hassan et al., 2019; Zhang et al., 2020b;a; Hassan et al., 2021b; Huang et al., 2022; Zhao et al., 2022) or generating short-term, predefined motions, such as reaching or sitting, given a static object (Starke et al., 2019; Taheri et al., 2020; Zhang et al.,

2021; 2022; Taheri et al., 2022). More recent work (Wang et al., 2021a;b; Hassan et al., 2021a; Lee & Joo, 2023; Jiang et al., 2024a; Cha et al., 2025; Kim et al., 2025b) has extended this to synthesizing human motions in 3D scenes with multiple objects, while other work has explored human motion generation involving dynamic objects (Li et al., 2023a; Xu et al., 2023; Ghosh et al., 2023; Li et al., 2023b; Xu et al., 2024; Jiang et al., 2024b; Xu et al., 2025). These methods leverage 3D motion-scene paired datasets (Savva et al., 2016; Monszpart et al., 2019; Hassan et al., 2019; Yi et al., 2024; Kim et al., 2025c) or 3D human-object interaction datasets (Taheri et al., 2020; Bhatnagar et al., 2022; Jiang et al., 2023; Wang et al., 2023b; Kim et al., 2023a; Xie et al., 2024; Baik et al., 2025) to enable such scene-conditioned motion generation. To address the scarcity of large-scale 3D datasets, some approaches (Han & Joo, 2023; Kim et al., 2024a; Li & Dai, 2024; Kim et al., 2024b) leverage the knowledge of 2D generative or vision language models, yet are limited to static human-scene interactions. In this work, we synthesize 3D HOI motions from the 2D scene, utilizing our target-aware video diffusion model.

## 3 PRELIMINARIES: VIDEO DIFFUSION MODELS

Building on the success of text-to-image latent diffusion models (Rombach et al., 2022b; Black Forest Labs, 2023; Esser et al., 2024), recent text-to-video (T2V) diffusion models (Blattmann et al., 2023; Yang et al., 2025b; HaCohen et al., 2024) generate videos in a latent space. Given a video $x$, an encoder $\mathcal{E}$ maps it to its latent representation $z$. During the forward process, Gaussian noise is added to $z$ at each timestep $t$, as $z_t = \alpha_t z + \sigma_t \epsilon$, where $\epsilon \sim \mathcal{N}(0, I)$ and $\alpha_t$, $\sigma_t$ are noise scheduling coefficients. In the reverse process, the model is trained to predict the noise added to the video latent, guided by input conditions such as text prompt, by minimizing the following objective:

$$\mathcal{L}_{\text{VDM}} = \mathbb{E}\left[\|\epsilon - \epsilon_\theta(z_t; y, t)\|_2^2\right], \tag{1}$$

where $y$ denotes the text prompt. T2V diffusion models can be fine-tuned for image-to-video (I2V) tasks by conditioning the model on an extra input image, allowing the video to start from the given image (Xing et al., 2024; Yang et al., 2025b; Kong et al., 2024; HaCohen et al., 2024). In this work, we use CogVideoX (Yang et al., 2025b), one of the SOTA open-sourced video diffusion models based on diffusion transformers (Peebles & Xie, 2023).

## 4 TARGET-AWARE VIDEO DIFFUSION MODELS

Given an image, a mask of the target, and a text prompt describing an action, our target-aware video diffusion model generates videos where an actor accurately interacts with the specified target. We first extend our base video diffusion model to accept the mask as an additional input (Sec. 4.1). To make the model utilize the extra mask information, we augment the text prompt by adding a sentence, "The person interacts with [TGT] object.", where the [TGT] token is used to encode the spatial information of the target. We then apply a cross-attention loss to align the cross-attention maps of the [TGT] token with the mask input and make our model target-aware (Sec. 4.2). This loss is selectively applied to specific cross-attention regions and transformer blocks to maximize its effectiveness (Sec. 4.3). Finally, we curate a dedicated dataset tailored to train our target-aware model (Sec. 4.4).

### 4.1 SPECIFYING THE TARGET WITH A MASK

To encode the spatial information of the target, we extend our base I2V diffusion model (Yang et al., 2025b) to incorporate a binary segmentation mask of the target. Our base model takes an input image $I \in \mathbb{R}^{H \times W \times 3}$ and generates an output video $V \in \mathbb{R}^{F \times H \times W \times 3}$, where $F$ denotes the number of frames. To enforce the input image as the first frame of the video, the latent noise $z_t \in \mathbb{R}^{f \times h \times w \times c}$ is concatenated channel-wise with the latent encoding of the input image $\mathcal{E}(I) \in \mathbb{R}^{1 \times h \times w \times c}$ for the first frame, with zero-padding applied for the remaining frames. The concatenated representation is then projected through an image projection layer to align with the text embedding dimension.

To integrate the target mask $M \in \mathbb{R}^{H \times W \times 1}$ into this process, we downsample it to $\tilde{M} \in \mathbb{R}^{h \times w \times 1}$ and concatenate it alongside the input image condition, again applying zero-padding for the other frames. To support the extra mask channel, we extend the original image projection layer by adding an input channel, and initialize the new weights to zero while preserving the pretrained parameters, following InstructPix2Pix (Brooks et al., 2023). The overall pipeline is shown in Fig. 1a.

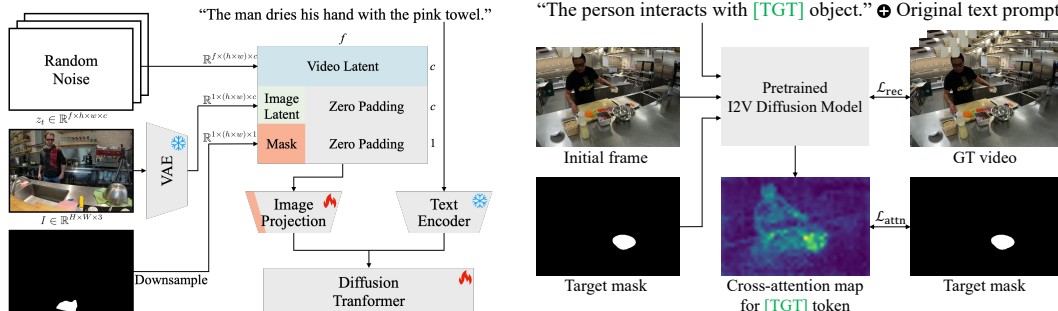

(a) Injecting the extra mask condition       (b) Target awareness via cross-attention loss

Figure 1: **Target-aware video diffusion models.** (a) We condition the noisy video latent with a segmentation mask of the target to incorporate spatial information during generation. (b) We fine-tune the pretrained video diffusion model to utilize the mask input via additional cross-attention loss.

## 4.2 TARGET AWARENESS VIA CROSS-ATTENTION LOSS

We make our model target-aware by applying a cross-attention loss that aligns the model's attention on the target with the additional input mask during fine-tuning. For every text prompt in our training dataset, we append a general sentence "The person interacts with [TGT] object.", where the [TGT] token is intended to encode the target's spatial information. We then encourage the cross-attention weights between the latent noise corresponding to the first frame of the video and the [TGT] token to align with the provided target mask $M$ as demonstrated in Fig. 1b. Specifically, we minimize the following loss:

$$\mathcal{L}_{\text{attn}} = \mathbb{E}\left[\|A(\boldsymbol{z}_t^0, [\text{TGT}]) - \tilde{\boldsymbol{M}}\|_2^2\right], \tag{2}$$

where $A(\boldsymbol{z}_t^0, [\text{TGT}])$ denotes the cross-attention weights between the latent noise for the first frame of the video and the [TGT] token. In addition to the cross-attention loss, we employ the standard diffusion objective:

$$\mathcal{L}_{\text{rec}} = \mathbb{E}\left[\|\boldsymbol{\epsilon} - \boldsymbol{\epsilon}_\theta(\boldsymbol{z}_t; \boldsymbol{y}, \boldsymbol{I}, \tilde{\boldsymbol{M}}, t)\|_2^2\right], \tag{3}$$

where notations remain consistent with those used in Eq. (1). Our overall objective is defined as:

$$\mathcal{L}_{\text{total}} = \mathcal{L}_{\text{rec}} + \lambda_{attn}\mathcal{L}_{\text{attn}}, \tag{4}$$

where $\lambda_{attn}$ balances the two loss terms. During inference, we prepend the [TGT] token to words referring to the target, enabling the model to leverage the spatial cue provided by the segmentation mask. As presented in Fig. 2a, the cross-attention loss effectively guides the [TGT] token to focus on the target region, enabling precise interactions with it.

## 4.3 SELECTIVE CROSS-ATTENTION LOSS

For effective and efficient supervision, we selectively apply the cross-attention loss to the model by identifying (1) the cross-attention regions that most influence target awareness of the model and (2) the transformer blocks that best capture semantics. We validate these design choices via ablations.

**Selective Cross-Attention Regions.** The multi-modal diffusion transformer architecture, employed by state-of-the-art image and video diffusion models (Black Forest Labs, 2023; Esser et al., 2024; Kong et al., 2024), including our base model (Yang et al., 2025b), concatenates text and video embeddings into a unified sequence and computes attention over the combined representation. This process yields four distinct attention regions: text-to-text self-attention, text-to-video (T2V) cross-attention, video-to-text (V2T) cross-attention, and video-to-video self-attention. While both T2V and V2T cross-attention maps encode semantic information (Fig. 2b top row), we apply the cross-attention loss on V2T cross-attention regions to maximize its impact since V2T cross-attention directly influences the video latent representations during the dot product computation of the attention weights and value features. In contrast, T2V cross-attention primarily affects the text latents, offering less direct influence on the video content. Details on attention mechanisms are provided in Appendix Sec. C.4.

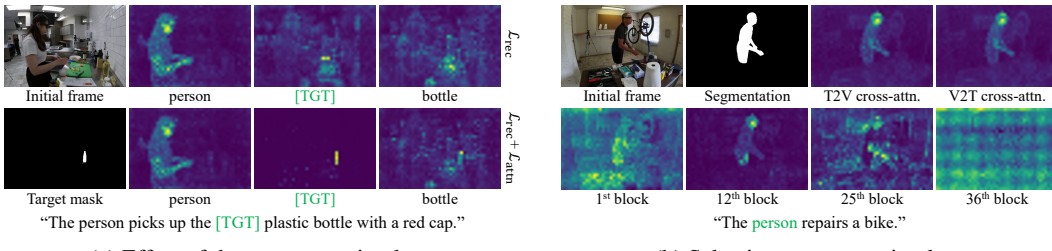

(a) Effect of the cross-attention loss        (b) Selective cross-attention loss

Figure 2: **Cross-attention visualization.** (a) The cross-attention loss successfully guides the model to focus on the target region. (b) We apply the loss on transformer blocks and cross-attention areas that largely impact target awareness of the model.

**Selective Transformer Blocks.** Motivated by prior work (Hertz et al., 2022), we observe that certain transformer blocks capture richer semantic details than others, as shown in the bottom row of Fig. 2b. We therefore apply the cross-attention loss to those blocks whose cross-attention maps closely resemble the segmentation masks of the corresponding token. To empirically identify these blocks, we evaluate the semantic alignment of each transformer block by first generating 100 videos from a subset of our training images. We then compute the mean squared error between each block's cross-attention map and the segmentation mask for a predefined token. Our analysis shows that blocks 5 through 23 of the base model (Yang et al., 2025b) yield the smallest errors, and we consequently apply the loss to uniformly sampled blocks (every $5^{th}$) within this range, constrained by GPU VRAM limitations.

### 4.4 DATASET CURATION FOR TRAINING THE TARGET-AWARE MODEL

To train our target-aware model, we require videos that satisfy two conditions: (1) the initial frame should depict a scene where an actor is present but not yet interacting with the target, and (2) subsequent frames must capture the actor engaging with the target. Directly collecting such data through one's own captures is infeasible, since it would not provide sufficient scale or diversity. To this end, we curate a dedicated dataset for target-aware video generation, designed to cover diverse interaction scenarios and align with our training pipeline. Each video, sourced from BEHAVE (Bhatnagar et al., 2022) and Ego-Exo4D (Grauman et al., 2024) datasets, is annotated with a segmentation mask of the target in the initial frame and paired with text prompts describing the action. BEHAVE dataset features videos where a single person interacts with a clearly defined target object in a relatively simple setting, whereas Ego-Exo4D contains more complex scenarios, such as cooking or bike repairing, where multiple objects, including those of the same type, may be present. In total, we extract 1290 clips that meet our criteria. We obtain the mask for the target object in the initial frame using an off-the-shelf segmentation model (Kirillov et al., 2023) and generate text prompts with CogVLM2-Caption (Yang et al., 2025b), the same captioning tool used for training our base model. While it is ideal to prepend [TGT] tokens to the target object nouns during caption generation, we find that current video captioning tools cannot reliably identify the target object in complex scenes. Therefore, we add a general sentence, "The person interacts with [TGT] object." to the generated captions as described in Sec. 4.2, which we find sufficient to train our target-aware model.

## 5 PRACTICAL APPLICATIONS OF OUR TARGET-AWARE MODEL

The core strength of our target-aware model lies in its ability to generate plausible and diverse interaction motions between actors and specified target objects, without requiring additional guidance. Leveraging this capability, we present two practical pipelines: (1) zero-shot 3D HOI motion synthesis for a target object with physical plausibility and (2) long-term video content creation with minimal user input. See the detailed pipelines in Appendix Sec. A.2.

### 5.1 ZERO-SHOT 3D HOI MOTION SYNTHESIS WITH PHYSICAL PLAUSIBILITY

Given an actor in a 2D scene and a desired target object, our model produces realistic HOI actions aligned with text prompts, providing strong planning cues for robotics control (Du et al., 2023; Ajay et al., 2023; Black et al., 2024; Ni et al., 2024; Park et al., 2026). To validate this connection, we present a pipeline that first applies an off-the-shelf 3D human pose estimator (Shen et al., 2024)

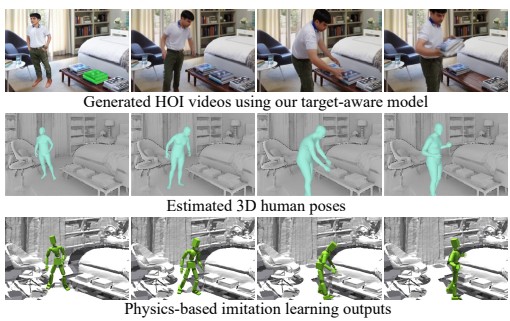

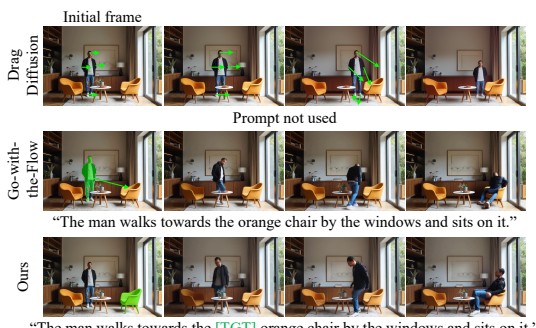

Figure 3: **Zero-shot 3D HOI motion synthesis.** We perform imitation learning on 3D poses of a person interacting with a target in the scene, obtained from videos generated with our model.

Figure 4: **Comparison over drag-based methods.** Ours with less extensive inputs outperforms drag-based editing methods (Shi et al., 2024b; Burgert et al., 2025).

to videos generated by our model, extracting 3D human motion sequences. We then perform a physics-based imitation learning (Wang et al., 2023b) to train a policy that mimics these motions in the Isaac Gym simulator (Makoviychuk et al., 2021). As shown in Fig. 3, the resulting agents reproduce human-object interactions in a physically plausible manner, demonstrating the potential of our approach to bridge between video generation and robotics.

## 5.2 LONG-TERM VIDEO GENERATION WITH TARGET-AWARE INTERACTIONS

Our target-aware model serves as a key component for video content creation, enabling effective control over an actor's actions in a scene without extensive manual effort. We introduce a simple yet robust pipeline that combines a video interpolation technique between keyframes and our target-aware video diffusion model to support two types of actions: navigating the scene and interacting with target objects. For navigation, we interpolate between two keyframes using an off-the-shelf frame interpolation model (Fei, Zhengcong, 2024). Each keyframe is constructed by placing the actor at a desired location in the scene via our depth-aware 3D insertion method. To generate HOI actions with a specified target object, we first position the actor using the same insertion method, specify the target with an off-the-shelf segmentation tool (Kirillov et al., 2023), and finally employ our target-aware model to synthesize realistic interactions. Importantly, our target-aware model provides a convenient way to produce plausible HOI scenes by simply specifying a target, which can then be connected through interpolation-based models for long-form video synthesis. The overall pipeline is illustrated in Fig. 10 of Appendix Sec. A.2.

## 6 EXPERIMENTS

We are the first to introduce a target-aware video diffusion framework that explicitly models actor-target interactions. To rigorously evaluate this new task, we construct a dedicated benchmark, introduce metrics, and establish strong baselines for comparison.

### 6.1 EXPERIMENTAL SETUP

**Dataset.** We construct a benchmark set of 80 images depicting scenes with a person, where each image is paired with a text prompt describing an interaction between the person and a target object. For all pairs, we ensure that the target can be clearly distinguished with text prompts using a noun, a color descriptor, or a spatial detail (e.g., soda bottle on the table, blue box at the center). Text prompts follow the format "The person {action} with {object}." for baselines and "The person {action} with [TGT] {object}." for ours. These prompts are further refined using GPT-4o, following the prompt enhancement procedure of CogVideoX (Yang et al., 2025b). For each test image, we generate 5 videos with random seeds, comparing results with a total of 400 samples.

**Metric.** We evaluate our approach along two dimensions: target alignment and generation quality. To measure target alignment, we assess whether the generated video captures an accurate interaction between the person and the target object. Specifically, we employ an off-the-shelf contact detector (Narasimhaswamy et al., 2020) to identify human-object contact in each frame of the generated

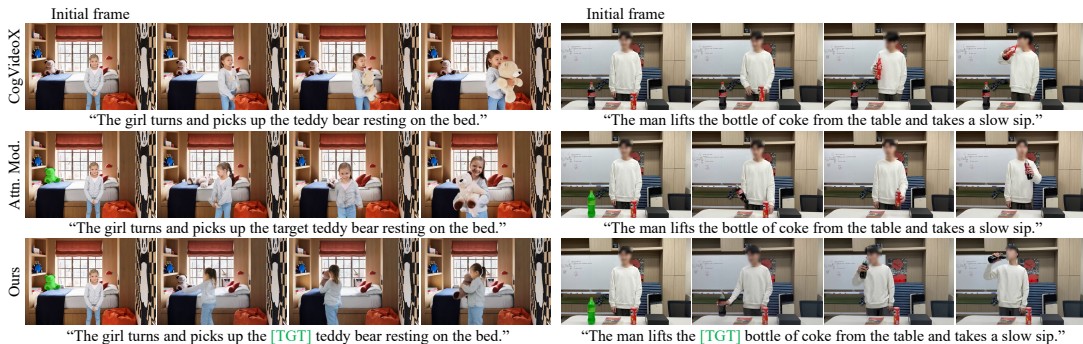

Figure 5: **Qualitative comparison on target alignment.** Each set displays generated videos using different methods. While baselines tend to hallucinate the target, our target-aware model produces videos where the actor interacts accurately with the actual target in the scene.

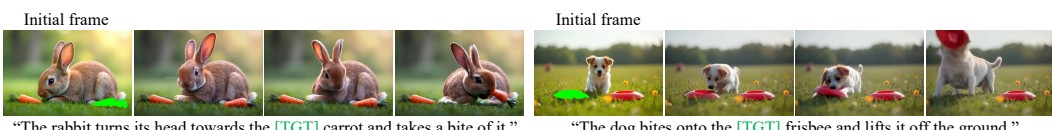

Figure 6: **Non-human interactions.** Our target-aware model generalizes to non-human cases despite being fine-tuned on human-scene interaction videos.

video and consider the interaction accurate if the detected contact regions overlap the target object's mask in at least one frame. We report the rate of accurate interactions over all generated videos (*Contact Score*). In addition, we perform two types of user studies (*Hum. Eval.* and *User Pref.*) to further assess the target alignment of each method. For generation quality, we adopt the evaluation metrics from VBench (Huang et al., 2024), which break down video quality into subject consistency (*SS*), background consistency (*BC*), dynamic degree (*DD*), motion smoothness (*MS*), aesthetic quality (*AQ*), and imaging quality (*IQ*). The final score (*Avg.*) is computed by averaging them.

**Baselines.** Since our method uses a single mask to specify the target, direct comparisons with state-of-the-art approaches that rely on heavy temporal conditioning on the actors are not appropriate. Instead, we evaluate against three representative baselines. First, we evaluate against our base image-to-video diffusion model, vanilla CogVideoX (Yang et al., 2025b). Second, we assess against a version of CogVideoX fine-tuned on videos of our dataset to isolate the effect of our method from that of the additional data (*CogVideoX w. data*). Finally, we compare with the attention modulation method from Direct-a-video (Yang et al., 2024), which enforces a subject's trajectory by amplifying cross-attention weights within predefined bounding box regions (*Attn. Mod.*). Since we assume that trajectory annotations for actors and targets are unavailable in our evaluation setting, we adapt this method by prepending the keyword "target" to the object description in the prompt and amplifying cross-attention weights in the target object mask region for that keyword.

## 6.2 QUALITATIVE EVALUATION

**Target Alignment.** Fig. 5 compares our method with baselines in terms of targeting accuracy. In rows 1 and 2, baseline methods occasionally hallucinate the target described in the prompt, rather than incorporating the actual target from the input image. In contrast, our approach generates videos where the actor accurately interacts with the specified target.

**Multiple Objects of the Same Type.** Fig. 7 highlights our key advantage where the scene contains multiple target objects of the same type. By enabling the usage of an explicit segmentation mask to identify the target, our method ensures precise selection and manipulation of the intended target.

**Non-Human Interactions.** While our model is fine-tuned on human-scene interactions, it generalizes well to interactions involving non-human subjects, as shown in Fig. 6.

**Specifying Both the Actor and the Target.** Our approach enables simultaneous control over both the source actor and the target object, as demonstrated in Fig. 8. We extend our model to accept two separate segmentation masks as additional inputs and introduce two tokens, [SRC] and [TGT]. Each token is encouraged to attend to each mask with our cross-attention loss during fine-tuning. During inference, we prepend each token to the actor and target descriptions, respectively.

| | Targeting Quality | | | Video Quality | | | | | | |
|---|---|---|---|---|---|---|---|---|---|---|
| | Contact Score↑ | Hum. Eval.↑ | User Pref.↑ | SC↑ | BC↑ | DD↑ | MS↑ | AQ↑ | IQ↑ | Avg.↑ |
| CogVideoX | 0.560 | 0.456 | 28.4% | 0.893 | 0.898 | 0.883 | 0.988 | 0.502 | 0.694 | 0.810 |
| CogVideoX w.data | 0.638 | 0.596 | 36.2% | 0.914 | 0.907 | 0.907 | 0.990 | 0.492 | 0.653 | 0.810 |
| Attn. Mod. | 0.546 | 0.508 | 22.2% | 0.872 | 0.889 | 0.786 | 0.986 | 0.499 | 0.687 | 0.786 |
| Ours | **0.878** | **0.892** | (100%-above) | 0.933 | 0.919 | 0.899 | 0.937 | 0.496 | 0.656 | 0.807 |

Table 1: **Quantitative comparison.** Our method enables the generation of videos containing accurate interactions with the specified targets. We also report generation quality with measures from VBench (Huang et al., 2024), confirming that our approach does not compromise video quality.

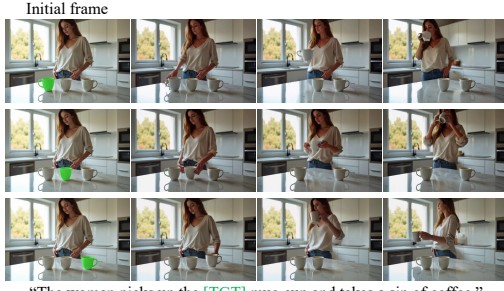

Initial frame    Initial frame

"The woman picks up the [TGT] mug cup and takes a sip of coffee."    "The [SRC] robotic arm picks up the [TGT] blue can with its robot hand."

Figure 7: **Multiple objects of the same type.** Our method ensures accurate interaction with the intended target by leveraging its mask.

Figure 8: **Control over multiple entities.** Our model can be extended to specify both the source actor and the target object using two masks.

**Comparison with Drag-Based Methods.** Fig. 4 shows a qualitative comparison with two drag-based image/video editing methods, which require additional user inputs to directly control the actor. DragDiffusion (Shi et al., 2024b) adjusts image latents based on drag operations. Since a single large drag is ineffective, we gradually move the actor toward the target using multiple small drags. While DragDiffusion produces reasonable results for small translations, it fails for larger adjustments. Go-with-the-Flow (Burgert et al., 2025) controls motion by warping the initial noise sequence to follow a desired flow, which we implement by dragging the actor's segmentation mask toward the target. Although this method enables the actor to make contact with the target, the output video lacks plausible motion due to its coarse conditioning. In contrast, our approach produces realistic interactions even without explicit motion guidance.

## 6.3 QUANTITATIVE EVALUATION

**Target Alignment and Video Quality.** As presented in Tab. 1 *Contact Score*, our method substantially outperforms all baselines in generating accurate interactions with target objects. At the same time, ours maintains video generation quality, achieving comparable scores to baselines across the *Video Quality* metrics in Tab. 1. The attention modulation approach fails to maintain the temporal consistency of videos since the amplified cross-attention values adversely affect the self-attention values, resulting in low contact scores. Additional details are provided in Appendix Sec. C.4.

**User Study.** We conduct two types of user studies via CloudResearch Connect. In (*Hum. Eval.*), each generated video is presented together with the corresponding input image and target object specification. Participants then make a binary judgment on whether the actor interacts accurately with the specified target. A total of 50 participants evaluate 10 videos per method, and we report the overall rate of accurate interactions in Tab. 1. In (*User Pref.*), each input image is presented alongside two generated videos: one produced by our method and the other by a baseline. Participants are asked to choose which video better reflects accurate interaction with the target. Again, 50 participants answer 10 questions per baseline, and we report in Tab. 1 the proportion of times each baseline is preferred over ours. In both studies, participants consistently favor our outputs by a large margin.

|  | Contact Score↑ | Video Quality↑ |
|---|---|---|
| Random | 0.819 | 0.807 |
| Equally-Spaced | 0.816 | 0.800 |
| Ours | **0.878** | 0.807 |

Table 2: **Cross-attention loss on selective transformer blocks.** We apply the loss to blocks that best capture semantics.

|  | Contact Score↑ | Video Quality↑ |
|---|---|---|
| T2V Cross-Attn. | 0.740 | 0.806 |
| Both Cross-Attn. | 0.860 | 0.810 |
| Ours (V2T Cross-Attn.) | **0.878** | 0.807 |

Table 3: **Cross-attention loss on selective attention regions.** We apply the loss to the attention regions that most influence target awareness.

|  | Contact Score↑ | Video Quality↑ |
|---|---|---|
| $\lambda_{attn} = 0.0$ | 0.647 | 0.815 |
| $\lambda_{attn} = 0.05$ | 0.727 | 0.811 |
| $\lambda_{attn} = 0.1$ | 0.878 | 0.807 |
| $\lambda_{attn} = 0.25$ | **0.890** | 0.806 |
| $\lambda_{attn} = 0.5$ | 0.888 | 0.807 |
| $\lambda_{attn} = 1.0$ | 0.888 | 0.804 |

Table 4: **Effects of different cross-attention loss weights.** Incorporating our cross-attention loss is crucial for achieving target awareness.

|  | Contact Score↑ | Video Quality↑ |
|---|---|---|
| original | 0.896 | 0.812 |
| dilate-3 | 0.884 | 0.808 |
| dilate-5 | 0.872 | 0.815 |
| erode-3 | **0.904** | 0.815 |
| erode-5 | 0.880 | 0.813 |

Table 5: **Effect of the mask quality.** Even when masks are expanded or shrunk, the Contact Score remains stable, supporting that our method is not sensitive to precise segmentation.

## 6.4 ABLATION STUDIES

**Cross-Attention Loss on Selective Blocks.** We evaluate the impact of applying cross-attention loss on different transformer blocks. For all experiments, we fix the number of blocks receiving the loss per training step to seven. We compare three strategies: (1) random seven blocks at each training step, (2) seven equally spaced blocks, and (3) blocks chosen using our proposed method. As shown in Tab. 2, our approach shows improved target alignments.

**Cross-Attention Loss on Selective Regions.** In Tab. 3, we examine how applying cross-attention loss on various regions of the cross-attention influences performance. The results indicate that the loss should be applied on the V2T cross-attention for better target alignment.

**Cross-Attention Loss Weight.** In Tab. 4, we analyze the impact of cross-attention loss coefficient $\lambda_{attn}$. When $\lambda_{attn} = 0.0$, meaning the model is trained solely with the reconstruction loss, the target alignment performance is nearly identical to that of the CogVideoX fine-tuned on our dataset (Tab. 1, second row). This demonstrates that simply introducing the mask does not, by itself, improve the target awareness of the model, and incorporating the cross-attention loss is essential. As we increase the loss weight, we observe a saturation of the contact score and set $\lambda_{attn} = 0.1$.

**Quality of Masks.** To evaluate the effect of segmentation quality on target alignment, we dilate and erode the masks at varying levels. As shown in Tab. 5, our method remains robust to these perturbations, as even coarse masks effectively guide the model by narrowing the region of interest.

**Shape of Masks.** We further assess the model's robustness to mask shape by replacing the original mask with a circular mask centered at the original mask's bounding box center, with a radius of 15 or 30 pixels. As demonstrated in Tab. 6, our model allows abstract spatial cues as input.

|  | Contact Score↑ | Video Quality↑ |
|---|---|---|
| original | **0.896** | 0.812 |
| circular-15 | 0.838 | 0.813 |
| circular-30 | 0.888 | 0.809 |

Table 6: **Effect of the mask shape.** Our method does not depend on the exact mask shape.

## 7 CONCLUSION

We presented a target-aware video diffusion model that generates videos where an actor plausibly interacts with a specified target, defined by a segmentation mask in the first frame. Our goal is to establish target awareness as a core capability of video generative models. Under this formulation, our model naturally guides the actor to interact with the designated targets in realistic and semantically consistent ways. Experiments demonstrate the strengths and advantages of our model compared to baseline methods and alternative solutions. Finally, we present key applications of zero-shot 3D HOI motion synthesis and video content creation using our target-aware video diffusion model.

## ACKNOWLEDGMENTS

This work was supported by RLWRLD, NRF grant funded by the Korean government (MSIT) [No. RS-2022-NR070498, No. RS-2025-25396144, and No. PJT-25-122310], and IITP grant funded by the Korea government (MSIT) [No. RS-2024-00439854, No. RS-2025-25442338, No. RS-2021-II211343, and No. RS-2025-02653113]. H. Joo is the corresponding author.

We thank Jeonghwan Kim for implementing physics-based imitation learning and rendering those results. We also thank Inhee Lee for proofreading the paper. H. Joo is the corresponding author.

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

# A    IMPLEMENTATION DETAILS

## A.1    TRAINING AND INFERENCE DETAILS

We use the CogVideoX-5B-I2V model (Yang et al., 2025b) as our base image-to-video diffusion model, producing output videos at a resolution of $720 \times 480$ with a total of 49 frames. For model fine-tuning, we employ LoRA (Hu et al., 2022) with a rank of 128 and $\alpha = 64$ to the diffusion transformer and designate the word "target" as our [TGT] token. We optimize the added LoRA layers and the extended image projection layer while keeping the other parts of the model frozen and train for 2,000 steps using an AdamW optimizer with a learning rate of $1 \times 10^{-4}$, and an effective batch size of 4. The addition of a single channel to the image projection layer for incorporating the target mask introduces 15,360 additional parameters to the 5B-parameter base model, resulting in negligible extra computational overhead. We set the cross-attention loss coefficient $\lambda_{attn} = 0.1$ and apply the loss to the video-to-text (V2T) cross-attention regions of every third transformer block from the 5th block to the 23rd block of the total 42 blocks. This selective application reduces VRAM usage by 71% compared to applying the loss across all blocks. We average the attention maps of these selected blocks and regions and normalize them to $[0, 1]$. The overall training takes approximately 6 hours on 4 NVIDIA A100 GPUs. For inference, we employ a DPM sampler (Lu et al., 2022) with $T = 50$ sampling steps and set the classifier-free guidance scale (Ho & Salimans, 2022) to 6 with the same dynamic guidance strategy as the original work (Yang et al., 2025b). The inference for a single video approximately takes 249.8 seconds on a single NVIDIA A100 GPU.

## A.2    DETAILS ON APPLICATIONS

**Application 1: Physics-Based Imitation Learning.** We use the official code of PhysHOI (Wang et al., 2023b) to implement physics-based imitation learning on 3D human poses extracted from our output videos. Since our goal is to learn a policy for human motion, we disable modules related to object motions during training. Joint training of full modules by obtaining paired data of 3D human pose and 3D object pose via an off-the-shelf object 6D pose estimator (Zhang et al., 2023a; Wen et al., 2024) could be a possible extension. Also, our current imitation learning outputs, as shown in Fig. 27, are manually aligned with the 3D scene due to different scales between estimated 3D human pose translations and the scene. Since the 3D location of the initial pose is given through 3D insertion of humans, we may adjust the scale of the subsequent translations, leveraging the depth information, which we leave as future work.

**Application 2: Inserting Humans into Scenes.**  As demonstrated in  Fig. 9, we perform human insertion in 3D space rather than in 2D pixel space to handle depth ordering and occlusions between the human and objects in the scene. Given an input human image, we use a single-view 3D human reconstruction method (AlBahar et al., 2023) to obtain a 3D reconstruction of the person. For the input scene image, we first apply a segmentation tool (Kirillov et al., 2023) to identify and segment the ground. We then use a metric-depth estimation method (Bochkovskii et al., 2025) to generate the real-scale 3D pointcloud. From the 3D pointcloud, we extract the points that belong to the ground when projected to the image and perform RANSAC-based plane fitting on these points to derive a 3D ground plane. Using the mapping between pixels and the 3D pointcloud, we obtain the 3D coordinates of the pixel to place the human. The reconstructed 3D human is positioned at its 3D point, perpendicular to the ground plane. To manage occlusions between the 3D human and the 3D scene pointcloud, we discard cases where significant overlap occurs and ask the user for alternative input coordinates. Once occlusion handling is complete, we render them together to obtain the human-inserted scene images.

Compared to 2D-based solutions using inpainting, where a specific region of the scene is masked and the person is inserted via personalized diffusion models (Rombach et al., 2022a; Ye et al., 2023), our 3D approach better preserves the appearance of the original inputs. As shown in  Fig. 11, inpainting-based methods often fail to maintain consistency with the original scene, resulting in undesired removal of objects in the scene. Additionally, inpainting can generate random details for occluded parts of the person in the input image, leading to inconsistencies between frames.

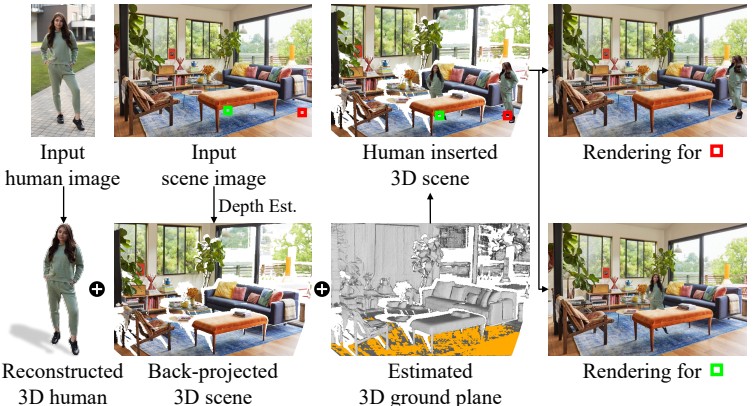

Figure 9: **Inserting humans into scenes.** We perform a 3D depth-based insertion of the human into the scene by performing single-view 3D reconstruction of the human and estimating the depth of the scene.

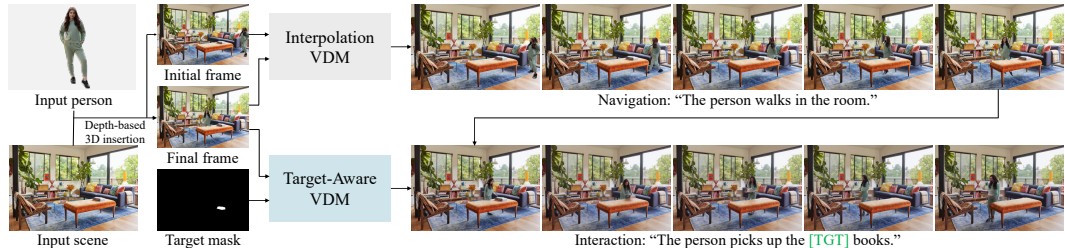

Figure 10: **Video content creation.** Given images of a person and a scene, we perform depth-based 3D insertion of the person into the scene and render them together to produce frames for video diffusion input. We interpolate generated initial and final frames to synthesize navigation contents, and utilize our target-aware video diffusion model to synthesize interaction contents.

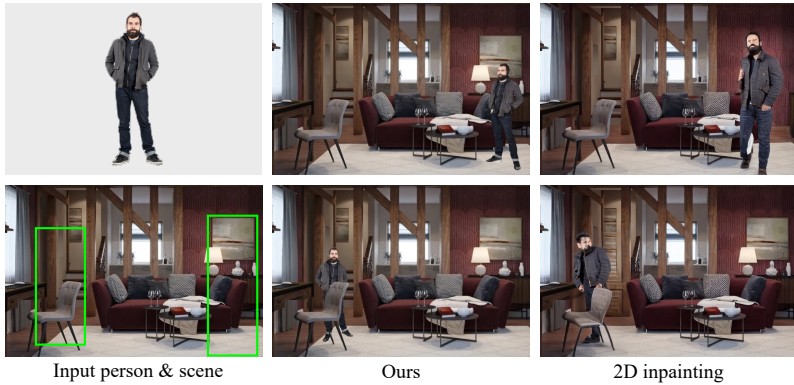

Figure 11: **Comparison with 2D inpainting.** Our 3D-based human insertion effectively inserts the human into the scene while preserving both identities and handling occlusion.

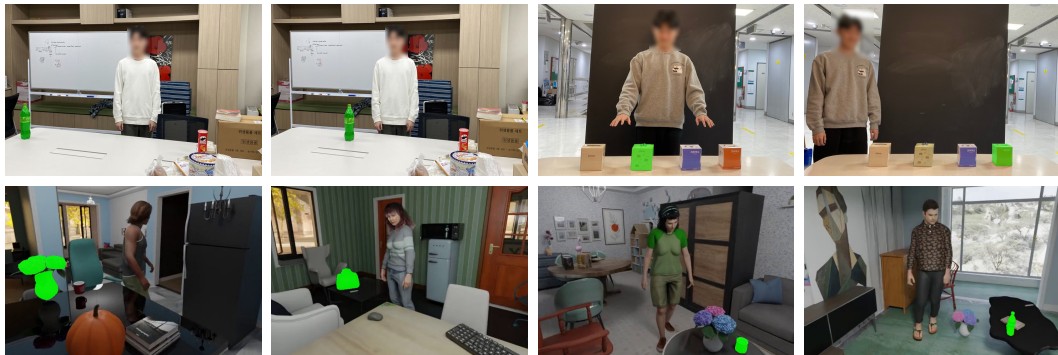

Figure 12: **Evaluation data example.** We show a subset of our evaluation dataset, where the target is indicated with a green mask. We confirm that the target is fully distinguishable with the input text prompt.

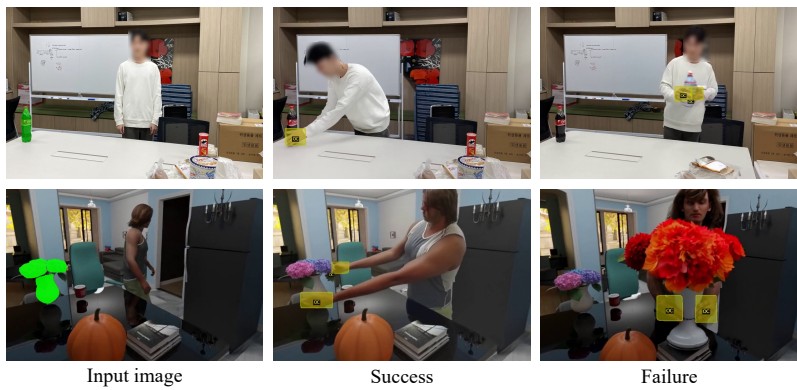

Input image               Success               Failure

Figure 13: **Contact score based on detection.** Green masks in the input image indicate the target, and yellow masks in output video frames indicate the detected object contact regions. We consider the interaction with the target accurate if the detected object contact region overlaps with the target mask.

### A.3 EVALUATION DETAILS

**Evaluation Dataset.** We construct a set of images of a scene containing a person paired with the prompt depicting an interaction between the person and the target. In the prompt, the person is described to interact with (1) an object placed at different locations of the scene relative to the person's position, or (2) a specific object among several different objects in the scene. For all images, we ensure that the target can be precisely determined with text prompts by a noun, a color description, or a spatial description. To get the target mask, we first perform instance segmentation on the image using SAM (Kirillov et al., 2023), and manually select the masks that belong to the target. Some image and mask samples of our evaluation dataset are presented in Fig. 12.

**Contact Score.** To detect physical contact between the actor and objects, we use the official code of ContactHands (Narasimhaswamy et al., 2020). We set the hand detection threshold to 0.5 and the object contact detection threshold to 0.5. We consider the interaction between the actor and the target to be successful when the detected object contact region overlaps with the segmentation mask of the target. Detection results for success and failure cases are presented in Fig. 13, where the yellow masks indicate the detected object contact regions.

**User Study.** We conduct two types of user studies to validate our comparisons. First, we perform human evaluation on videos generated by each method to assess whether the depicted person accurately interacts with the target. Fifty participants are presented with 10 videos per method. In each video, we show an input image with the target object highlighted with a green mask for 2 seconds, followed by the generated video. A screenshot of the human evaluation interface is shown in

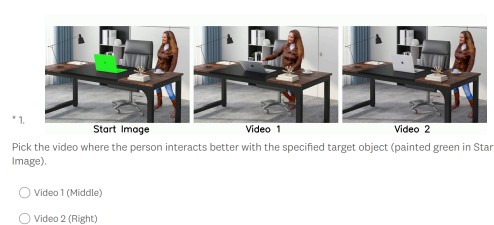

(a) Human evaluation on target alignment.  (b) User preference study on target alignment.

Figure 14: **User studies on target alignment.** (a) We ask participants to assess whether the person in the video accurately interacts with the target. (b) We ask participants to select the video that better demonstrates accurate target alignment.

Fig. 14a. We also perform an A/B test to measure user preference of our method over each baseline method in terms of target alignment. Fifty participants are asked 10 questions per baseline method, where each question displays an input image with the target in a green mask, alongside the generated videos from our method and a baseline in a random order. The screenshot of the user preference study interface is shown in Fig. 14b.

## B    ADDITIONAL QUALITATIVE RESULTS

**Non-human Interactions.** Fig. 15 is an extended figure of Fig. 6 in the main paper, demonstrating that our model generalizes to non-human interactions. Although the model is fine-tuned solely on a relatively small set of human-scene interaction videos and has never been exposed to actors other than humans during training, it can generate coherent and plausible non-human interactions. This generalization ability arises from the strong generative priors of the base video diffusion model, enabling the itself to adapt to novel agent types with target awareness.

**Targeting Objects in Outdoor Scenes.** Fig. 16 presents additional results of our target-aware model applied to outdoor scenes. Although the model is trained exclusively on indoor interaction videos, it generalizes well to diverse outdoor environments, successfully grounding the target and producing coherent actor-target interactions.

**Targeting Objects in Complex Scenes.** In  Fig. 17, we present additional results of our target-aware video diffusion model applied to complex scenes such as a bike repair shop or kitchen, where describing the target with text prompts is challenging. We also present the results of the original CogVideoX (Yang et al., 2025b). Our model successfully generates videos that capture accurate interactions with the target object, even when the target occupies only a small portion of a complex scene. Note that the scene, sourced from the Ego-Exo4D dataset (Grauman et al., 2024) is unseen during fine-tuning.

**Egocentric View Generation.** For robotics applications, generating videos from an egocentric perspective is particularly beneficial for capturing fine-grained actions. In Fig. 18, we demonstrate egocentric video generation using our approach. To better adapt the base model for this setting, we first fine-tune it on egocentric videos of the EgoIT-99 dataset (Yang et al., 2025a). We then apply our target-aware LoRA module to the fine-tuned model (CogVideoX-Ego) without any additional training. Notably, the target awareness generalizes seamlessly to the fine-tuned model.

**Providing Motion.** The output videos produced with our method, where the actor precisely interacts with the target, can serve as a source of motion data for existing controllable video generation approaches (Gu et al., 2025). Diffusion as Shader (Gu et al., 2025) uses 3D tracking video (Xiao et al., 2024) of a source clip to condition the motion in generated videos such that they follow the

Initial frame and mask

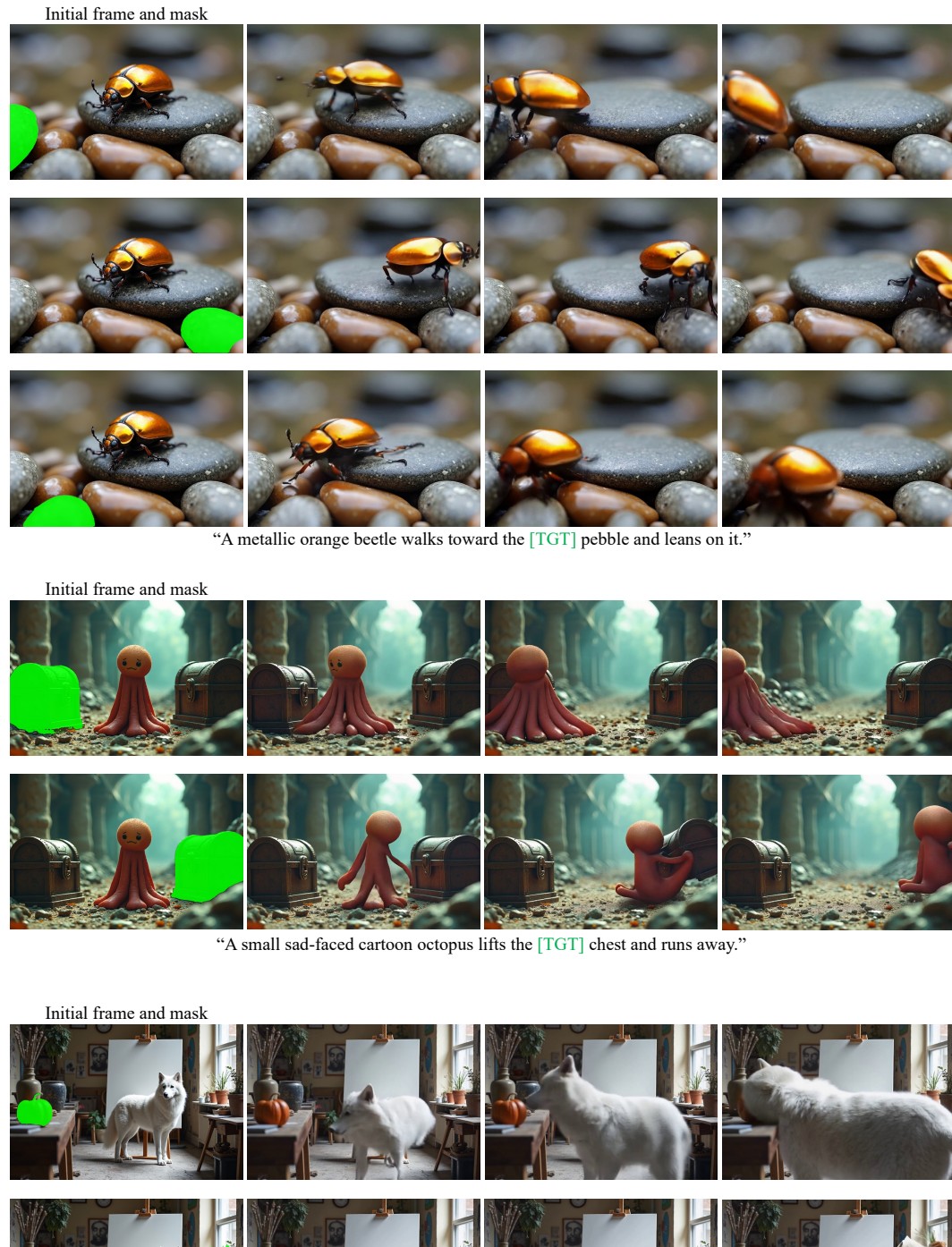

"A metallic orange beetle walks toward the [TGT] pebble and leans on it."

Initial frame and mask

"A small sad-faced cartoon octopus lifts the [TGT] chest and runs away."

Initial frame and mask

"A white wolf walks toward the [TGT] pumpkin/vase and bites it firm."

Figure 15: **Non-human interactions.** Our target-aware model generalizes well to non-human interactions despite being fine-tuned solely on 1K+ human-scene interaction videos.

Initial frame and mask

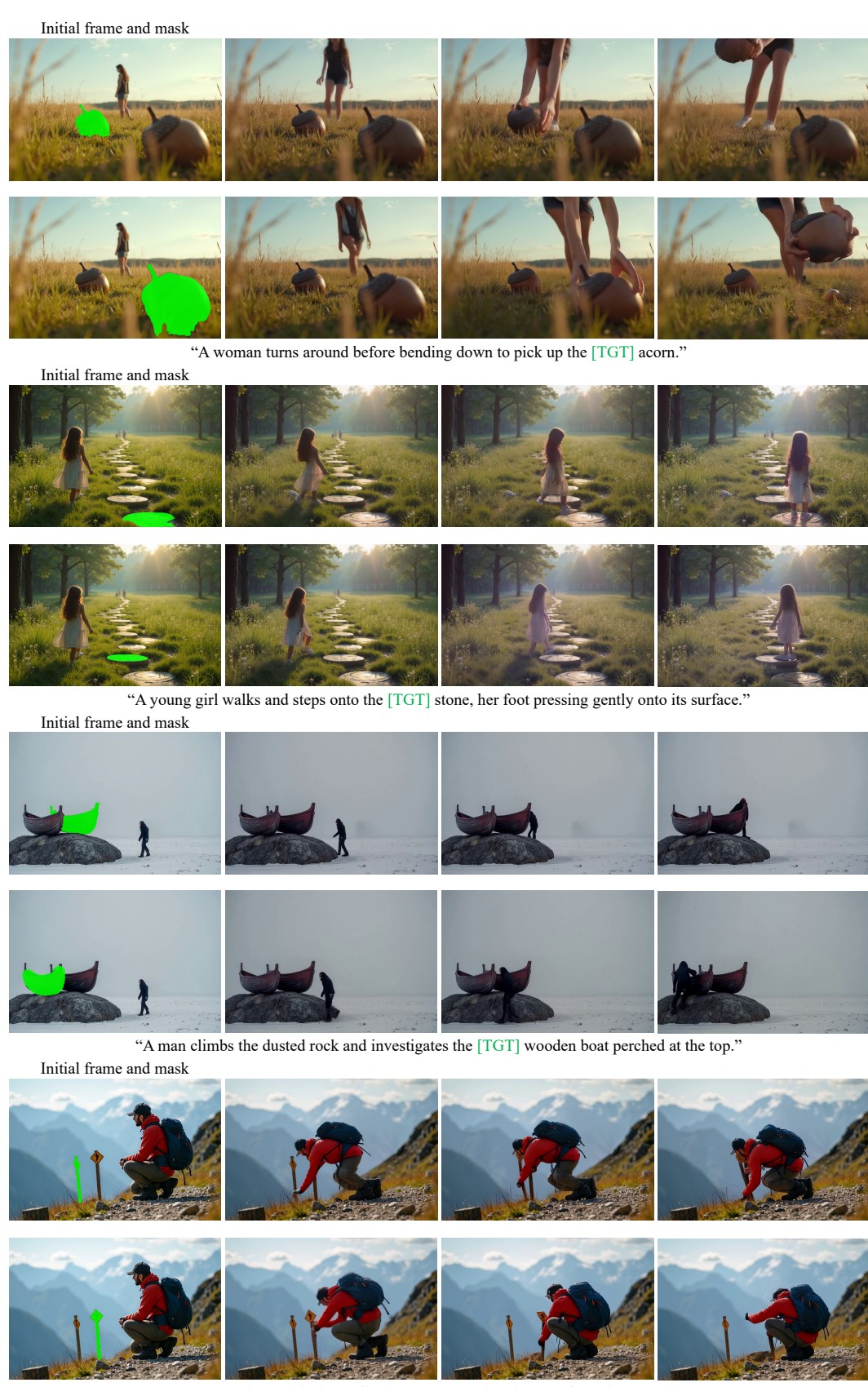

"A woman turns around before bending down to pick up the [TGT] acorn."

Initial frame and mask

"A young girl walks and steps onto the [TGT] stone, her foot pressing gently onto its surface."

Initial frame and mask

"A man climbs the dusted rock and investigates the [TGT] wooden boat perched at the top."

Initial frame and mask

"A man in a red jacket pulls the [TGT] wooden signpost from the ground."

Figure 16: **Target-aware generation in outdoor scenes.** Despite being fine-tuned only on indoor interaction videos, our model generalizes to interactions in diverse outdoor scenes.

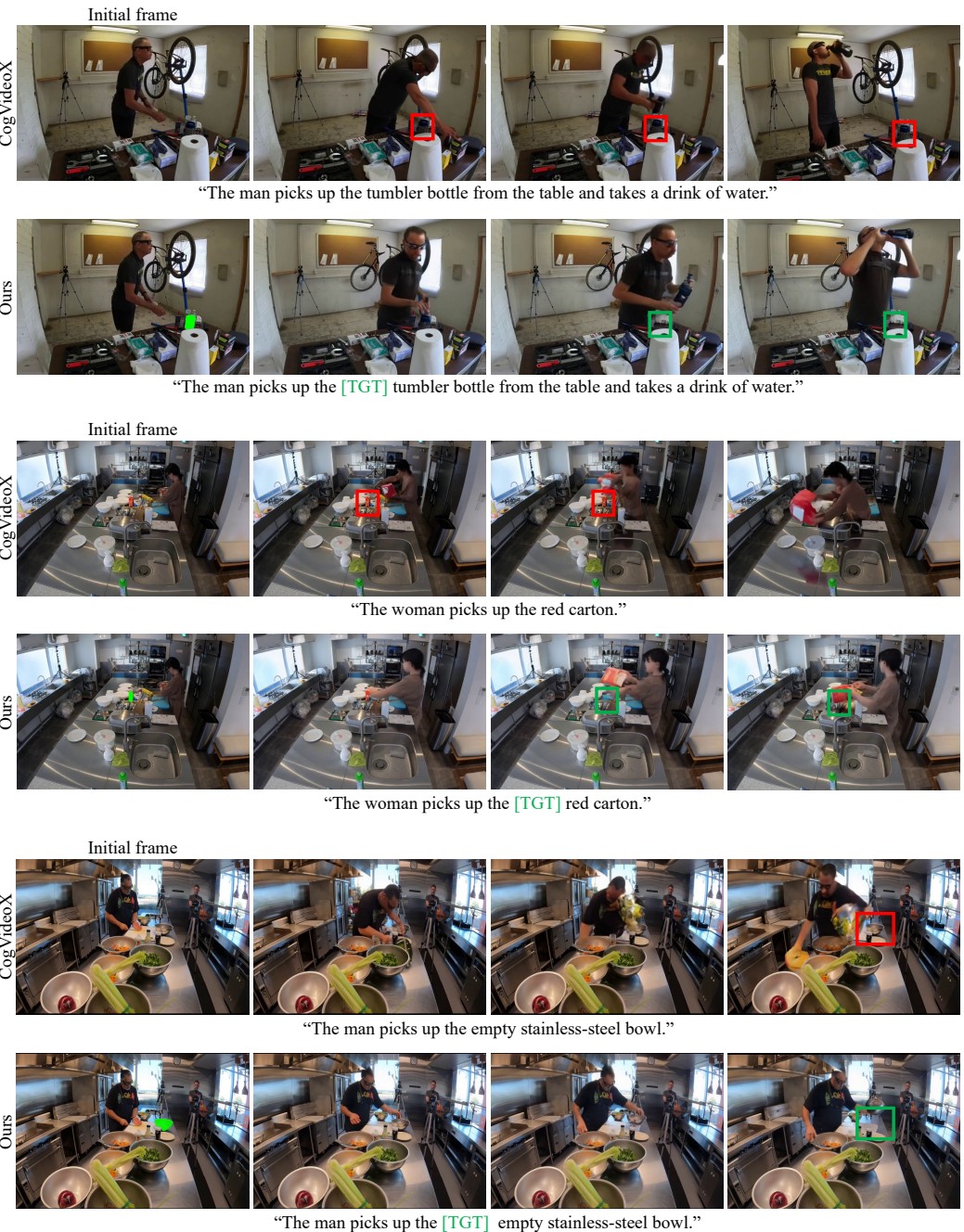

Figure 17: **Targeting objects in complex scenes.** We compare results of original CogVideoX (Yang et al., 2025b) and our target-aware model in complex scenes. Our model successfully generates target-aligned videos even when the target appears small in complex scenes. Best viewed with zoom.

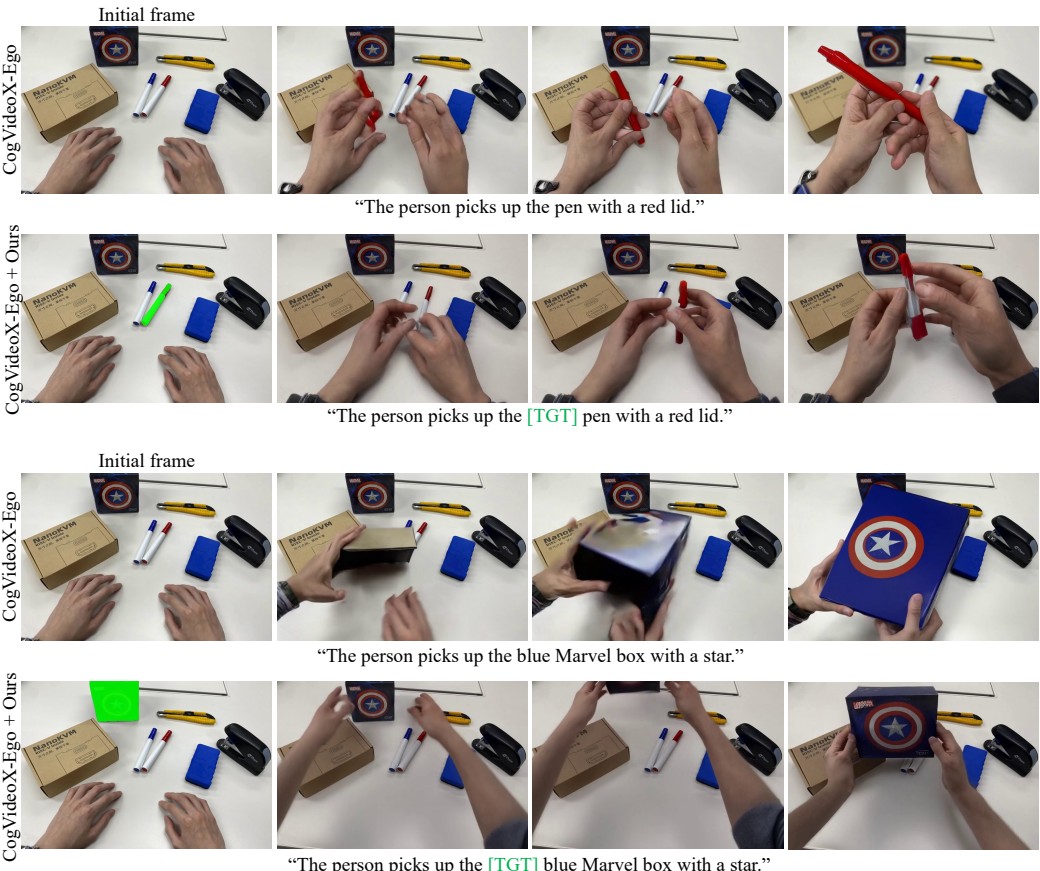

Figure 18: **Targeting objects in egocentric view.** We fine-tune the base model on egocentric videos and then apply our target-aware LoRA module. The target awareness seamlessly generalizes to the egocentric setting without additional training, enabling precise interactions with the specified object from the actor's viewpoint.

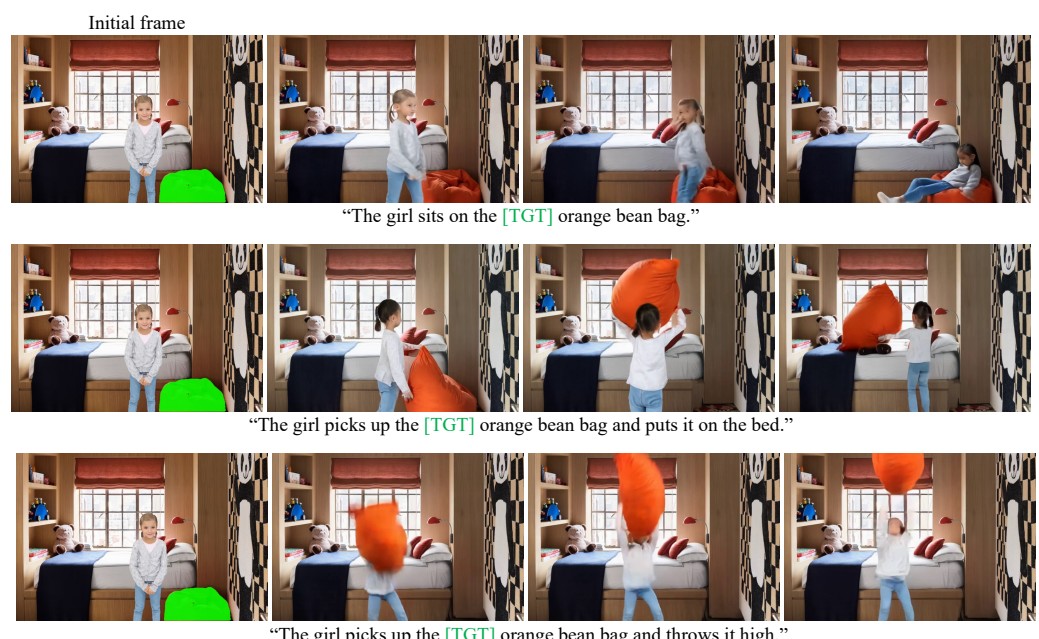

Figure 19: **Diverse actions with the same target.** Our method can generate diverse actions with the same target using different prompts.

Initial frame and mask

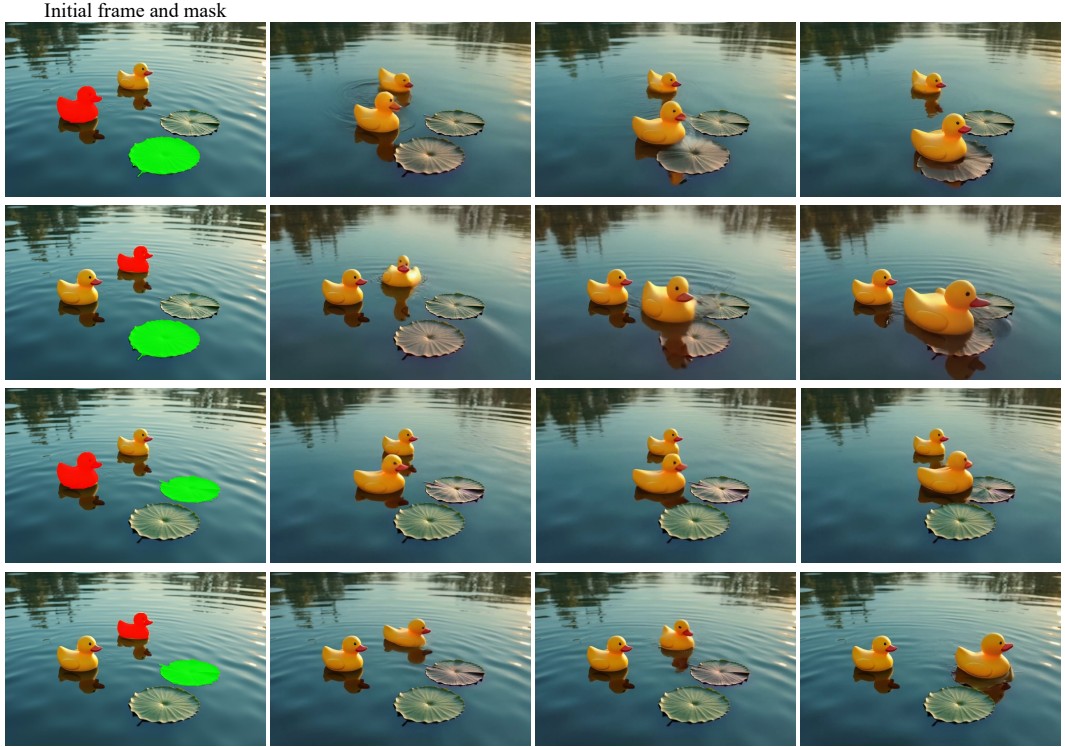

"The [SRC] rubber duck swims toward the [TGT] lily pad and sits on it."

Initial frame and mask

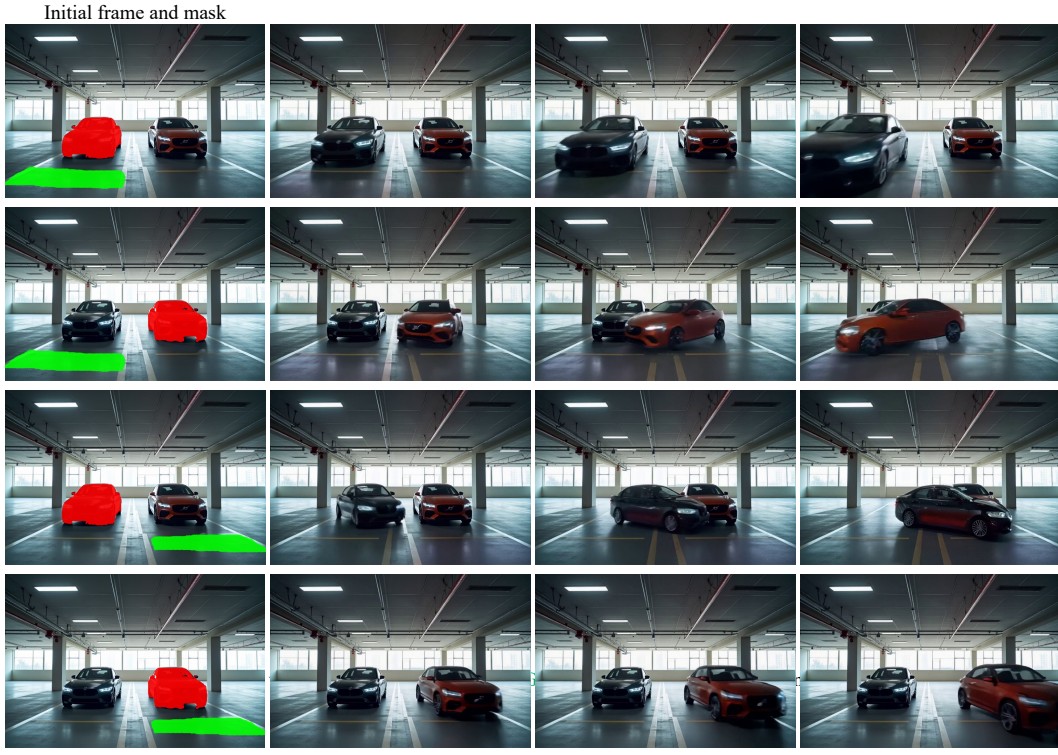

"The [SRC] car moves toward the [TGT] space."

Figure 20: **Control over multiple entities.** Our model can be extended to specify both the source actor and the target object using two masks.

Initial frame and mask

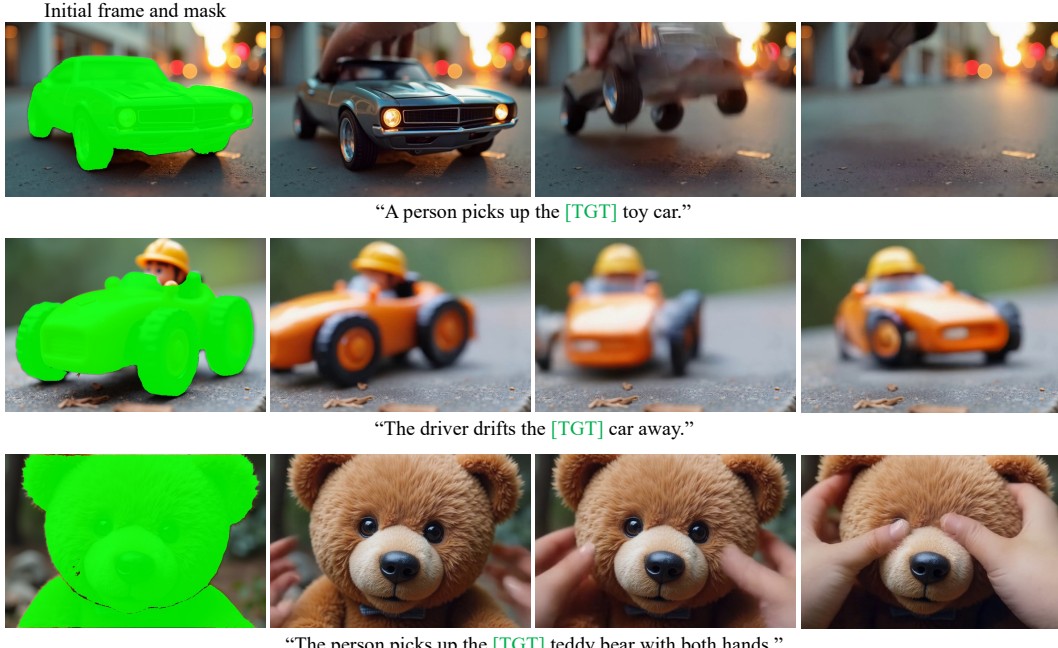

"A person picks up the [TGT] toy car."

"The driver drifts the [TGT] car away."

"The person picks up the [TGT] teddy bear with both hands."

Figure 21: **Large target in a scene.** Our method can handle targets that takes up a significant portion of the input frame.

motion of the source. However, acquiring appropriate source videos for complex motions, such as human-object interactions or robot manipulations, is often challenging. In such cases, our method can generate the desired interactions and provide sufficient motion conditions, as demonstrated in Fig. 26. We use Flux (Black Forest Labs, 2023) with a canny-edge ControlNet (Zhang et al., 2023b) to generate the initial frames for running Diffusion as Shader.

**Diverse Actions with the Same Target.** We demonstrate in Fig. 19 that our method can generate diverse actions with the same target by varying the prompt. The action quality depends on the base model, while the target awareness is applied via our method.

**Specifying Both the Actor and the Target.** Fig. 20 is an extended figure of Fig. 8 in the main paper. As demonstrated, our model can be extended to specify both the source actor and the target object using two masks for interaction.

**Scenes with Large Targets.** We demonstrate in Fig. 21 that our method can handle targets that takes up a large portion of the input frame. Our model continues to interpret the mask with the [TGT] specification correctly and generates interactions focused on the large target region.

**Applications.** Fig. 27 is an extended figure of Fig. 3 in the main paper, demonstrating the downstream applications of our method. Given images of a person and a scene, we first synthesize human-inserted images as described in Sec. A.2. As mentioned in the main paper, to achieve human navigation contents, we use a frame interpolation video diffusion model (Fei, Zhengcong, 2024) to interpolate two synthesized images where the person is inserted in different positions of the scene. For human action or manipulation content, we similarly start from a human-inserted image and utilize our model to achieve precise interaction with the target. From the generated contents, we extract the 3D human pose sequences (Shen et al., 2024) and use them to learn a policy via physics-based imitation learning (Wang et al., 2023b) given a target motion.

**Target Alignment.** Figs. 28 to 33 are extended figures of Fig. 5, demonstrating that our model enables accurate interactions between the actor and the target. We also provide generation outputs of the vanilla CogVideoX (Yang et al., 2025b) for comparison.

|  | Contact Score↑ | Video Quality↑ |
|---|---|---|
| Original | 0.896 | 0.812 |
| Automatic | 0.864 | 0.810 |

Table 7: **Automatic pipeline.** Our method stays robust even when target masks come from an automated pipeline, demonstrating that it does not rely on high-quality manual segmentation.

|  | Contact Score↑ | Video Quality↑ |
|---|---|---|
| General sentence only | 0.705 | 0.760 |
| General sentence with noisy descriptions | **0.878** | **0.807** |

Table 8: **Effect of removing captioner-generated descriptions during training.** Despite their noise, automatically generated captions help preserve the priors of the pre-trained backbone during fine-tuning.

## C  DISCUSSIONS

### C.1  MASKS

**Automatic Acquisition of Masks.** Although fully automating the inference pipeline is not the main focus of our work, we construct an automated pipeline to generate target masks from an input image and prompt, and evaluate its effectiveness. Specifically, we use GPT-4o (Hurst et al., 2024) to identify the target object noun from the input, and Grounded-SAM (Ren et al., 2024), an open-vocabulary segmentation tool, to obtain the corresponding mask. The pipeline runs in approximately 15 seconds on a single RTX 3090 and achieves 93.14% IoU accuracy on our evaluation set. We then use the automatically generated mask as input to our model and find that they achieve comparable performance, as presented in Tab. 7. For this experiment and the following mask ablations, we use a subset of 50 images from our evaluation set and similarly generate 5 videos per image.

### C.2  ROBUSTNESS TO NOISY CAPTIONS

**Training.** Our training dataset contains captions that may incorrectly describe which object the actor is interacting with, due to limitations of current video captioning models (Yang et al., 2025b). To handle this, we prepend a simple, but always true sentence, "The person interacts with [TGT] object." to the generated captions as mentioned in Sec. 4.4. This guarantees that the [TGT] token is semantically linked to the object under interaction, while the generated part of the caption mainly provides information about the actor's appearance, the scene, and coarse motions in the video. These descriptive details help preserve the priors of the pre-trained backbone during fine-tuning. To explicitly assess the role of these descriptive, but noisy details, we fine-tune the model using only the general sentence, "The person interacts with [TGT] object.", completely removing the captioner-generated descriptions. As demonstrated in Tab. 8, the variant trained with the general sentence only, maintains target awareness to some extent but exhibits degraded target alignment and video quality compared to the full setting with descriptions. This indicates that, despite their noise, automatically generated captions still contribute for the model to be target-aware while maintaining its priors.

**Inference.** We evaluate our model's robustness to noisy captions during inference by corrupting the textual descriptions while keeping the same target mask. In particular, we test (1) omitting the noun after [TGT], and (2) replacing the noun after [TGT] with an object name that does not correspond to the target. Qualitative results in Fig. 22 show that the model continues to interact with the masked target region under both perturbations. This further confirms that spatial grounding primarily arises from the [TGT] token and its alignment with the mask, rather than the specific noun used in the prompt.

### C.3  EXTENDED CONTACT SCORE

We propose *Contact Score* as our primary metric because it directly reflects the core objective of target-aware generation: whether the model correctly interacts with the specified target. In this

Initial frame and mask

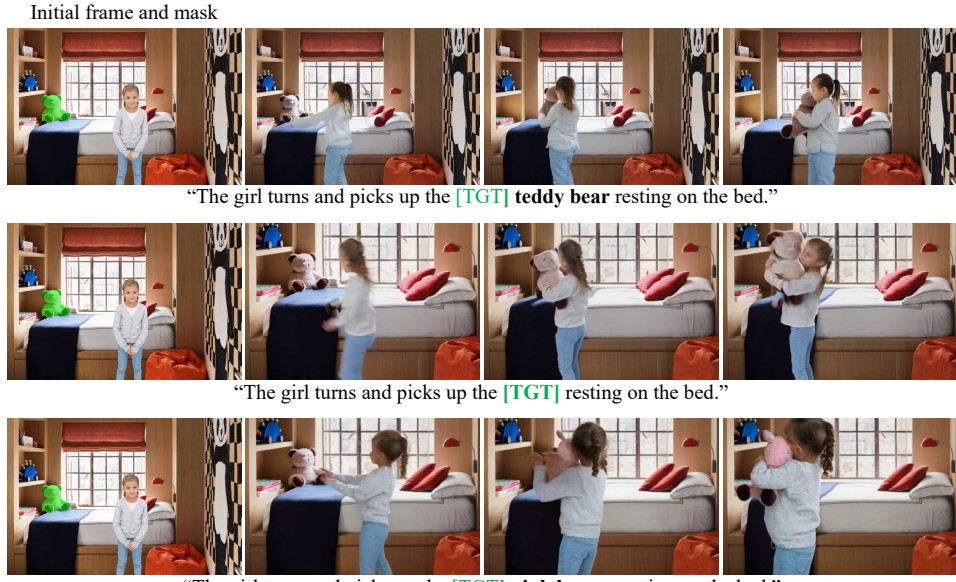

"The girl turns and picks up the [TGT] **teddy bear** resting on the bed."

"The girl turns and picks up the **[TGT]** resting on the bed."

"The girl turns and picks up the [TGT] **pink bunny** resting on the bed."

Figure 22: **Inference with perturbed prompts.** Our model continues to generate correct interactions with the specified target even when the noun following [TGT] is removed or replaced, demonstrating robustness to noisy captions at inference.

task setting, even brief contact with the correct target is more important than a long, semantically rich interaction with a hallucinated object. At the same time, single-frame overlap can be overly permissive in rare cases (e.g., accidental or spurious touches). To provide a more thorough evaluation, we introduce two complementary metrics.

We define *Contact Score (kf)* as a straightforward extension of Contact Score that counts an interaction as successful only if the detected contact region overlaps the target mask for at least $k$ consecutive frames (we report $k = 2$ and $k = 3$). This stricter criterion explicitly discounts accidental or transient touches. We note that *Contact Score (kf)* can sometimes underestimate correct interactions when the target moves outside its initial mask region within a few frames: in such cases, the contact may naturally leave the original target mask even though the interaction is correct.

We further introduce *Interaction Score* to require not only correct contact but also nontrivial target motion. Specifically, we track points sampled from the target-mask region of the first frame across the video using CoTracker (Karaev et al., 2024). We count an interaction as accurate only if (1) the original Contact Score condition holds (the contact region overlaps the target mask in at least one frame) and (2) the mean displacement of the tracked points exceeds a threshold (10 pixels in our experiments; static videos typically yield mean displacement below 0.5 pixels). This jointly enforces that the actor contacts the correct target and that the target undergoes meaningful motion. It is important to retain the original Contact Score because displacement alone can be triggered without correct interaction (e.g., target motion without being touched). A limitation of *Interaction Score* is that it can underestimate interactions where the target remains largely static (e.g., pushing on a fixed object or sliding a hand over a surface), since the displacement term remains small even when the interaction is correct. As demonstrated in Tab. 9, our method consistently outperforms baselines for all metrics.

## C.4 ATTENTION

**Attention Mechanisms in MM-DiTs.** State-of-the-art diffusion models (Black Forest Labs, 2023; Esser et al., 2024; Kong et al., 2024) including our base model (Yang et al., 2025b), utilize multi-modal diffusion transformers (MM-DiTs) (Esser et al., 2024) for denoising. In MM-DiTs, text and video latents are concatenated into a single sequence, and attention computations are performed over the combined representation. Specifically, given query features $\mathbf{Q}$, key features $\mathbf{K}$ and value features

| | Targeting Quality | | | |
|---|---|---|---|---|
| | Contact Score↑ | Contact Score (2f)↑ | Contact Score (3f)↑ | Interaction Score↑ |
| CogVideoX | 0.560 | 0.446 | 0.377 | 0.474 |
| CogVideoX w.data | 0.638 | 0.504 | 0.393 | 0.540 |
| Attn. Mod. | 0.546 | 0.399 | 0.340 | 0.455 |
| Ours | **0.878** | **0.770** | **0.693** | **0.783** |

Table 9: **Extended targeting metrics.** In addition to Contact Score, we report stricter temporal variants (Contact Score 2f/3f) and Interaction Score, which jointly requires correct target contact and nontrivial target motion.

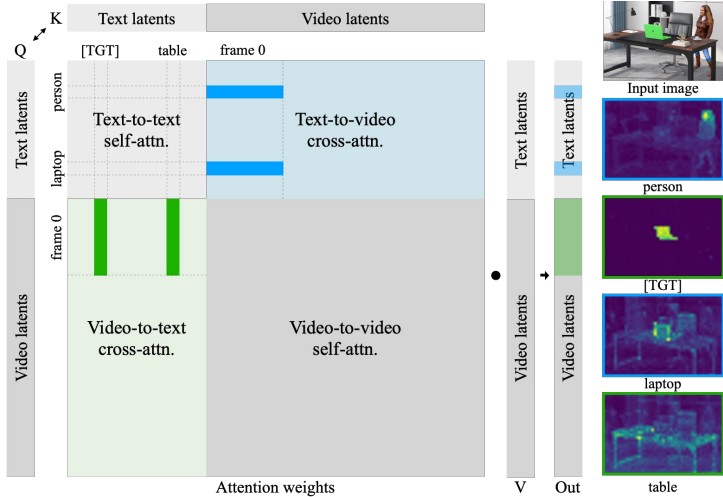

Figure 23: **Attention mechanisms in MM-DiTs.** Attentions of MM-DiTs can be divided into text-to-text self-attention, text-to-video (T2V) cross-attention, video-to-text (V2T) cross-attention, and video-to-video self-attention. Since V2T cross-attention weights directly influence the values of video latents, we apply our cross-attention loss on V2T cross-attention regions.

$\mathbf{V}$, each obtained by passing the combined representation through separate linear layers, the attention in a transformer block is computed as,

$$\text{Attn}(\mathbf{Q}, \mathbf{K}, \mathbf{V}) = \text{Softmax}(\frac{\mathbf{Q}\mathbf{K}^T}{\sqrt{d}})\mathbf{V}, \tag{5}$$

where $d$ is the channel dimension of $\mathbf{Q}$. The resulting $\text{Attn}(\mathbf{Q}, \mathbf{K}, \mathbf{V})$ is normalized, projected through linear layers, and used as the combined representation for the next transformer block. The attention weights are formed by $\text{Softmax}(\frac{\mathbf{Q}\mathbf{K}^T}{\sqrt{d}})$, where the value at index $[i, j]$ indicates the influence of the $i$-th token on the $j$-th token. As illustrated in Fig. 23, this process results in four distinct attention regions: text-to-text self-attention, text-to-video (T2V) cross-attention, video-to-text (V2T) cross-attention, and video-to-video self-attention.

As discussed in the main paper, while both T2V and V2T cross-attention maps encode semantic information, we find that applying our loss to the V2T cross-attention is more effective for enhancing target awareness. As demonstrated in Fig. 23, V2T cross-attention weights directly influence the video latents during the dot product computation of the attention weights and value features, whereas T2V cross-attention weights primarily affect the text latents. Although the influenced text latents can affect subsequent V2T cross-attention weights through $\mathbf{Q}\mathbf{K}^T$ computation, their impact is diminished, as shown in Tab. 3 of our main paper.

**Attention Modulation.** Prior work on controllable text-to-image generation (Hertz et al., 2022; Kim et al., 2023b; Ma et al., 2023; Chen et al., 2024; Xie et al., 2023) demonstrates that by modifying cross-attention maps during inference, it is possible to control the placement of subjects in specific

|  | $\lambda = 10$ | $\lambda = 25$ | $\lambda = 50$ | $\lambda = 100$ |
|---|---|---|---|---|
| $\tau = 0.80T$ | 0.493 | 0.473 | 0.527 | 0.513 |
| $\tau = 0.85T$ | 0.480 | 0.507 | 0.520 | 0.480 |
| $\tau = 0.90T$ | 0.533 | 0.520 | 0.573 | 0.520 |
| $\tau = 0.95T$ | 0.487 | 0.533 | **0.613** | 0.553 |

Table 10: **Contact scores for attention modulation.** We report contact scores for different combinations of attention control weights and cut-off timesteps.

regions of the output image. Cross-attention modulation is applied as follows:

$$\text{CrossAttnMod}(\mathbf{Q}, \mathbf{K}, \mathbf{V}) = \text{Softmax}(\frac{\mathbf{Q}\mathbf{K}^T + \lambda\mathbf{S}}{\sqrt{d}})\mathbf{V}, \tag{6}$$

where $\lambda$ denotes attention control weight, $\mathbf{S}$ is the modulation term with the same dimensions as the attention maps. The modulation term $\mathbf{S}$ takes positive values within the desired region for the subject and negative values outside that region. Formally, given a bounding box $\mathbf{B}$ that specifies the desired region for the object, $\mathbf{S}$ is defined as follows:

$$\mathbf{S}[i,j] = \begin{cases} 1 - \frac{\|\mathbf{B}\|}{|\mathbf{Q}\mathbf{K}^T|}, & \text{if } i \in \mathbf{B}, j \in \mathbf{P}, t \geq \tau \\ 0, & \text{if } i \in \mathbf{B}, j \in \mathbf{P}, t < \tau \\ -\infty, & \text{otherwise} \end{cases} \tag{7}$$

where $\|\mathbf{B}\|$ is the size of the bounding box, $|\mathbf{Q}\mathbf{K}^T|$ is the number of elements in $\mathbf{Q}\mathbf{K}^T$, $\mathbf{P}$ represents the indices of prompt tokens for subjects, and $\tau$ is a cut-off timestep. Since diffusion models form the subject layout in earlier steps (Xie et al., 2023; Hertz et al., 2022), the amplification is only applied in the early stage. Recently, the technique has been extended to text-to-video diffusion models, enabling control over object trajectories in generated videos by modulating attention map weights of every frame (Yang et al., 2024; Wu et al., 2024a).

As mentioned in the main paper, we adapt this attention modulation concept as a baseline. In our setting, since trajectories for actors and targets are not available, we add the word "target" to the object description in the prompt and amplify cross-attention weights in target mask regions for the new keyword. This modulation should mirror our approach without additional training. Since attention modulation modifies the internal attention computation during the denoising process, it is highly sensitive to hyperparameters such as the attention control weight $\lambda$ and the cut-off timestep $\tau$. We report the contact scores of each setting in Tab. 10, evaluated by generating three videos per image from our evaluation dataset. The results consistently show that the scores remain low even compared to the original CogVideoX (Yang et al., 2025b) without any modification. This degradation stems from the attention mechanisms in MM-DiTs: since the attention computation contains a row-wise softmax operation, modulating the cross-attention values affects the self-attention values of the video, ultimately leading to degraded output video quality and low contact scores, highlighting the necessity of our method for building target-aware models.

## C.5 LIMITATIONS AND FUTURE WORK

The quality of our generated videos is inherently constrained by existing open-sourced video models, which often produce noticeable visual artifacts when synthesizing complex appearances. Given that closed-sourced commercial models (Kli, 2024; Veo, 2024) yield more convincing results, we expect this limitation to be alleviated as more advanced open-sourced models become available. Nevertheless, enhancing video quality by incorporating interaction motion cues (Jeong et al., 2025; Chefer et al., 2025) could be an interesting future work.

Also, due to the static camera setting of our dataset, videos generated by our model tend to exhibit fixed camera trajectories. Given the scarcity of interaction datasets with dynamic cameras, integrating camera control techniques (Wang et al., 2024; Yang et al., 2024; He et al., 2025) into our model could be a possible future direction.

Another current limitation is that our architecture requires adding an extra channel per specified target, which poses scalability challenges when dealing with a large number of targets. Designing a

unified model that supports an arbitrary number of target specifications in a more memory-efficient manner presents an exciting future research direction.

Moreover, since our dataset contains masks that cover a single target object per video, our model mostly finds it difficult to handle cases where a single mask covers multiple objects (whether of the same or different categories), as demonstrated in Fig. 24. Robust multi-instance handling under a single mask remains future work.

Our current framework also assumes a fixed target specification throughout the whole video, which limits its ability to model complex interactions involving multiple objects over time (Fig. 25). A natural next step would be to support temporal target switching, enabling fine-grained control over which object is interacted with at each point in time. This would open new possibilities for long-horizon planning and action sequencing, especially in domains like robotics or instructional video generation.

Finally, our contact-based metrics rely on an off-the-shelf contact detector (Narasimhaswamy et al., 2020) that is designed specifically for human-object interactions and, in practice, only detects contacts involving human hands. As a result, these metrics cannot be directly applied to videos where the interacting agent is non-human (e.g., animals, robotic arms, tools). In such cases, the detector fails to produce meaningful contact predictions even when the generated interaction is qualitatively correct, leading to unreliable scores.

Initial frame

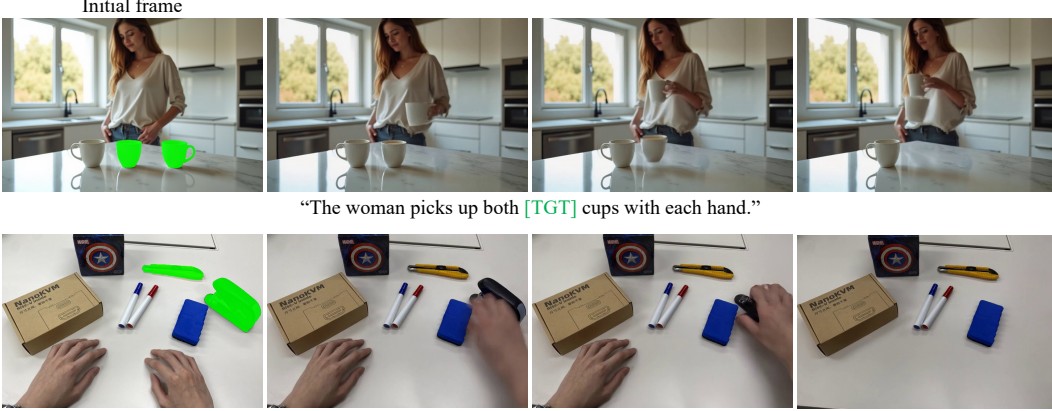

"The woman picks up both [TGT] cups with each hand."

"The person reaches out and picks up both [TGT] objects with each hand."

Figure 24: **Failure case: targeting multiple objects with a single mask.** Since our model is trained on masks that contains a single object, it struggles when a mask spans multiple objects.

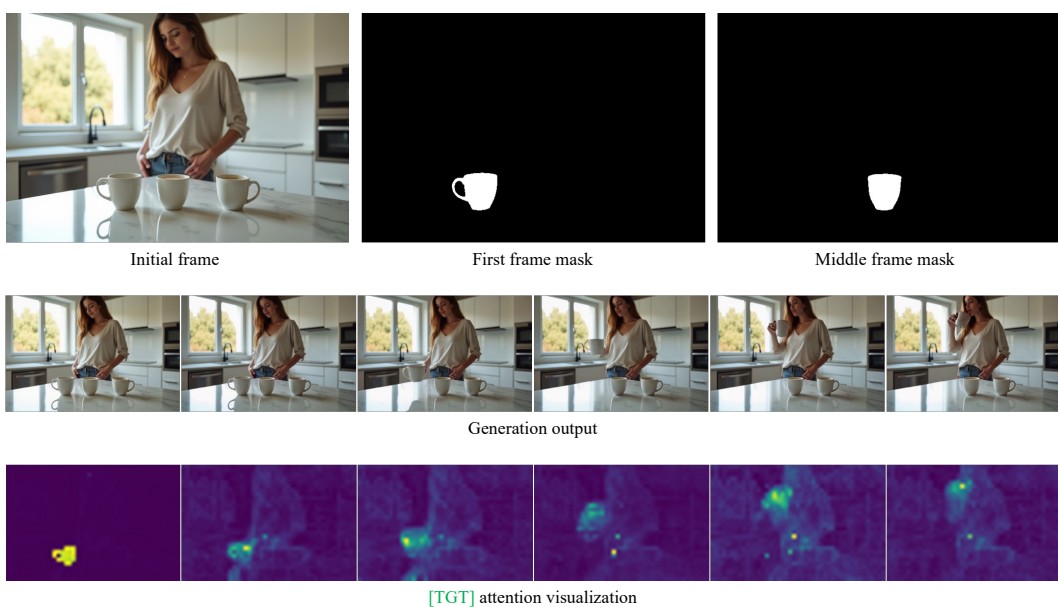

Initial frame      First frame mask      Middle frame mask

Generation output

[TGT] attention visualization

"The woman picks up the [TGT] cup and puts it back down. She then picks up the [TGT] cup and takes a sip of coffee."

Figure 25: **Failure case: targeting multiple objects over time.** Our framework assumes a single fixed target per generated video, preventing it from switching to new targets partway through the video. Even though an additional mask is provided at the middle frame, the model ignores it and continues to rely on the first-frame mask. As a result, it cannot handle sequential interactions with different targets within a single video. Nonetheless, target switching may still be achieved by generating a new video from the last frame of the previous one, enabling multi-target interactions through chained generation.

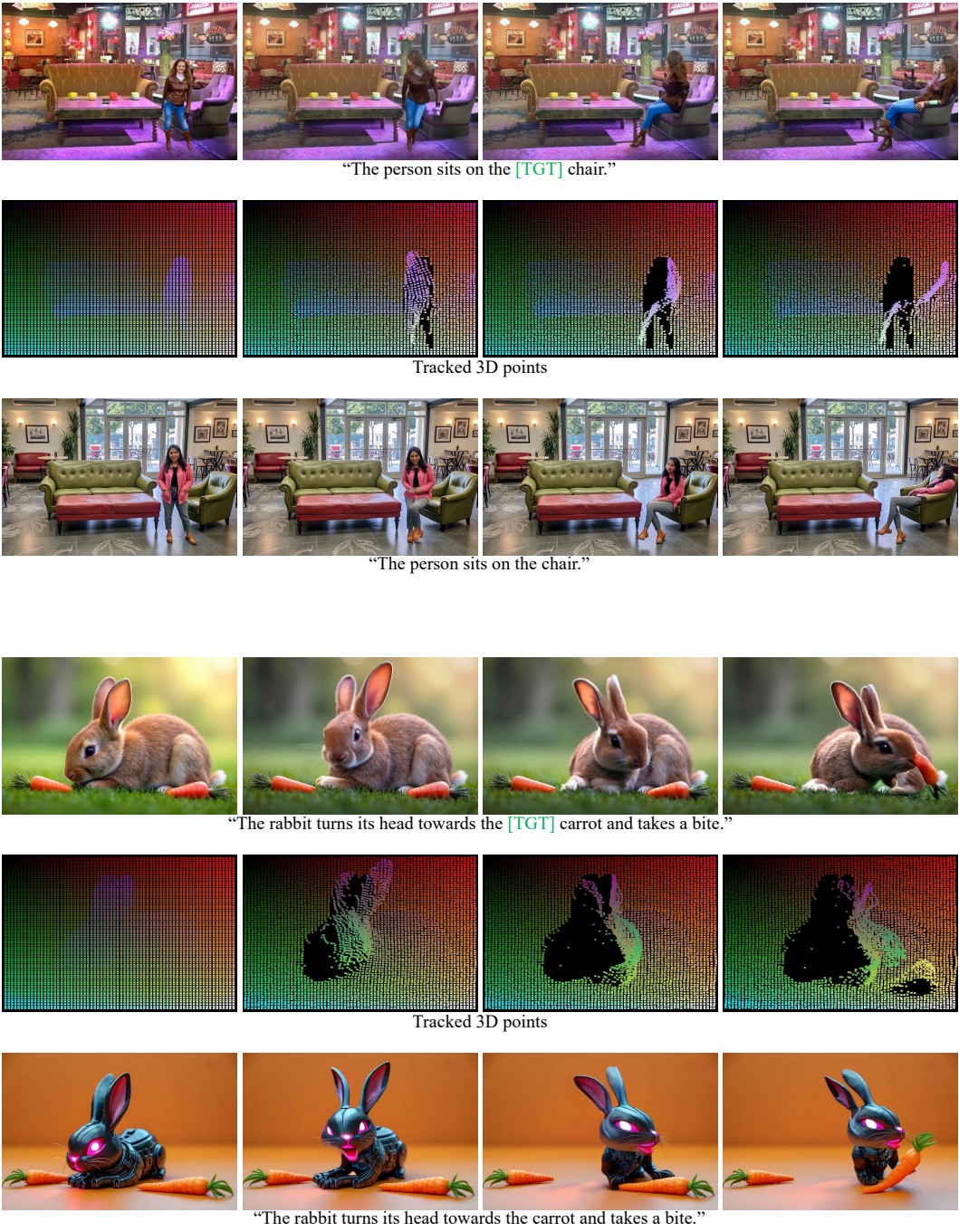

"The person sits on the [TGT] chair."

Tracked 3D points

"The person sits on the chair."

"The rabbit turns its head towards the [TGT] carrot and takes a bite."

Tracked 3D points

"The rabbit turns its head towards the carrot and takes a bite."

Figure 26: **Acting as a source video.** Our outputs can provide sufficient motion data for existing controllable video generation methods (Gu et al., 2025) that require dense structural conditions over frames.

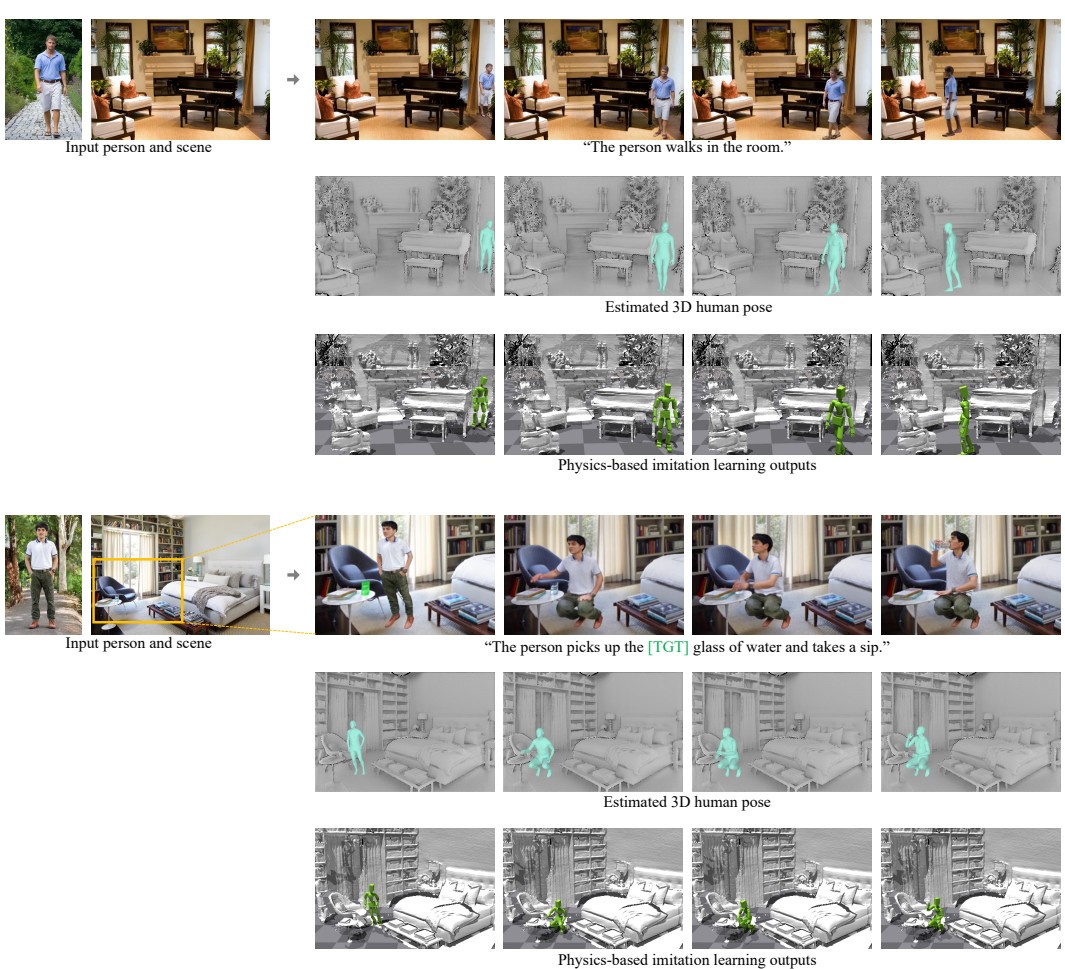

Figure 27: **Applications.** Given images of a person and a scene, we perform 3D insertion of the person into the scene and render them together to produce frames for video diffusion input. We interpolate synthesized initial and final frames to generate locomotion contents and utilize our Target-Specified video diffusion model to generate action and manipulation contents. We further demonstrate that extracted 3D human poses from our generated contents can be used as training data for physics-based imitation learning.

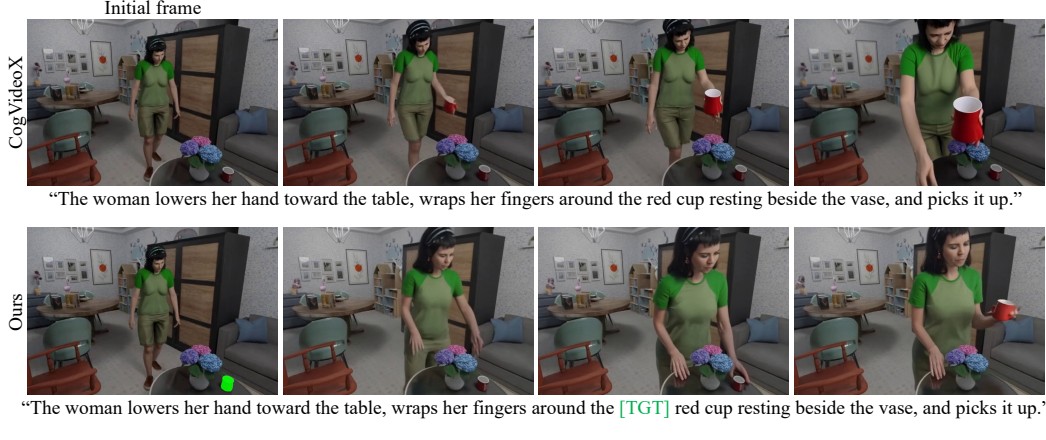

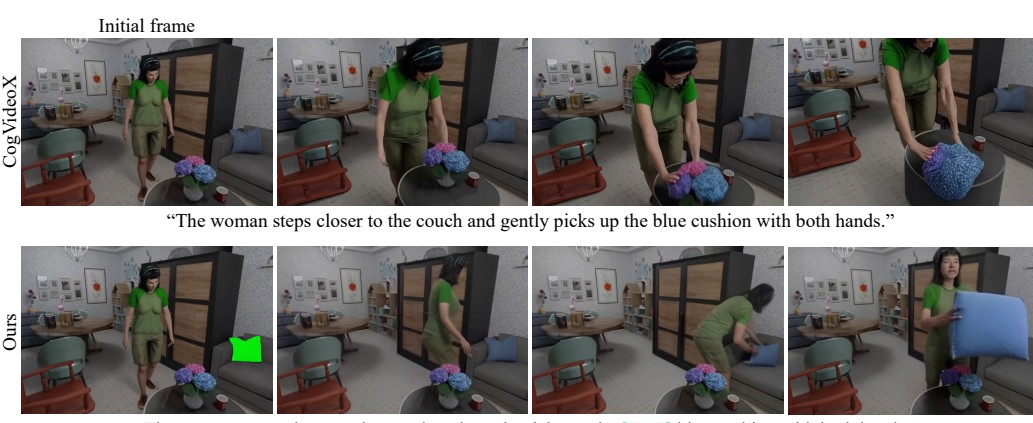

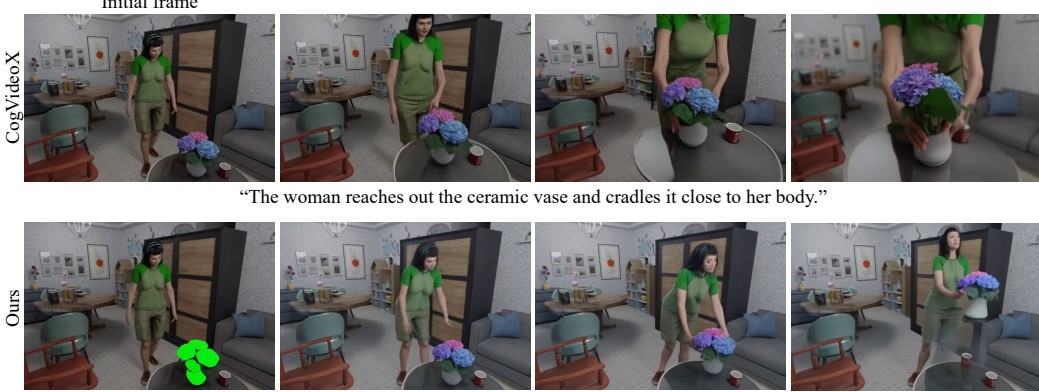

Figure 28: **Additional qualitative comparison on target alignment.** We compare results of original CogVideoX (Yang et al., 2025b) and our target-aware model. Our model successfully generates videos where the actor interacts accurately with the desired target. The target is colored in green every second row.

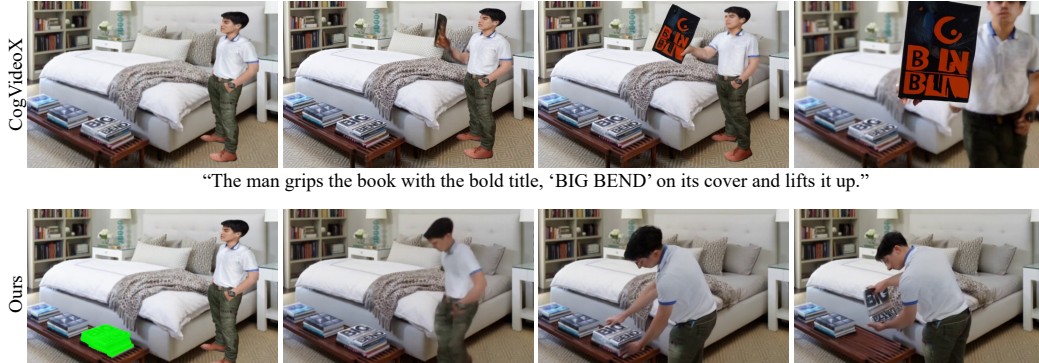

"The man grips the book with the bold title, 'BIG BEND' on its cover and lifts it up."

"The man grips the [TGT] book with the bold title, 'BIG BEND' on its cover and lifts it up."

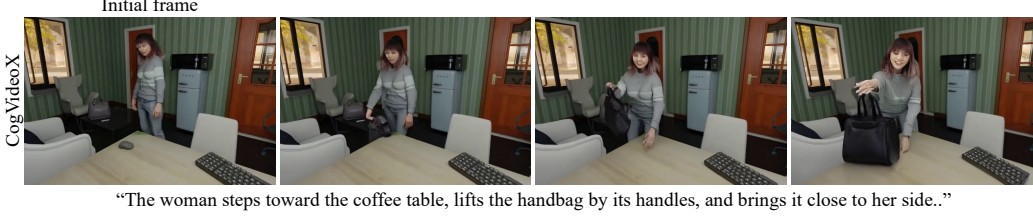

"The woman steps toward the coffee table, lifts the handbag by its handles, and brings it close to her side.."

"The woman steps toward the coffee table, lifts the [TGT] handbag by its handles, and brings it close to her side.."

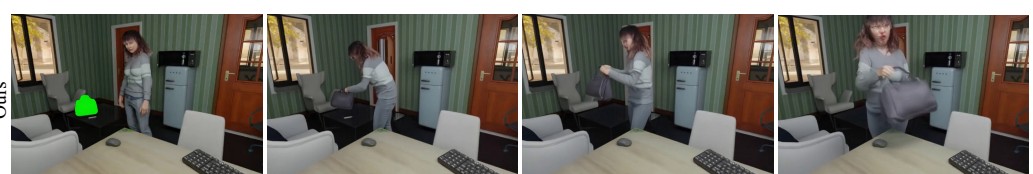

"The man leans forward, grasps the handbag placed beside his feet, and lifts it off the ground.."

"The man leans forward, grasps the [TGT] handbag placed beside his feet, and lifts it off the ground.."

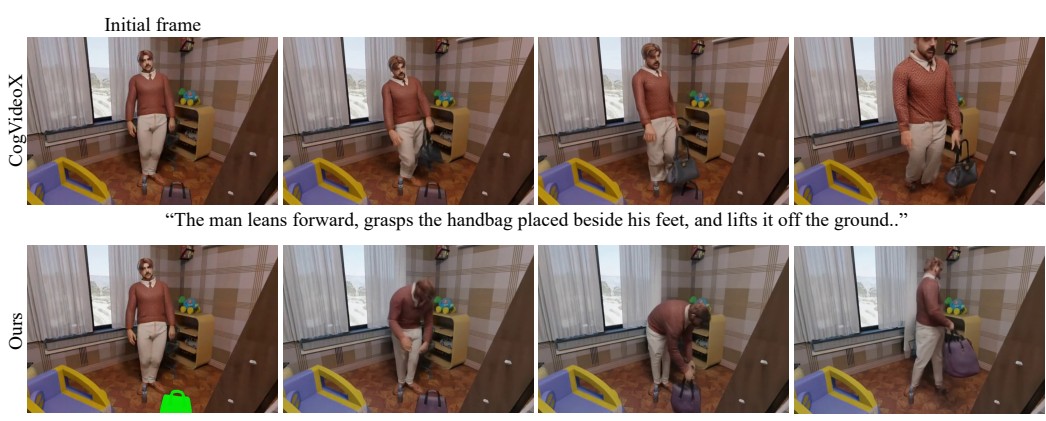

Figure 29: **Additional qualitative comparison on target alignment.** We compare results of original CogVideoX (Yang et al., 2025b) and our target-aware model. Our model successfully generates videos where the actor interacts accurately with the desired target. The target is colored in green every second row.

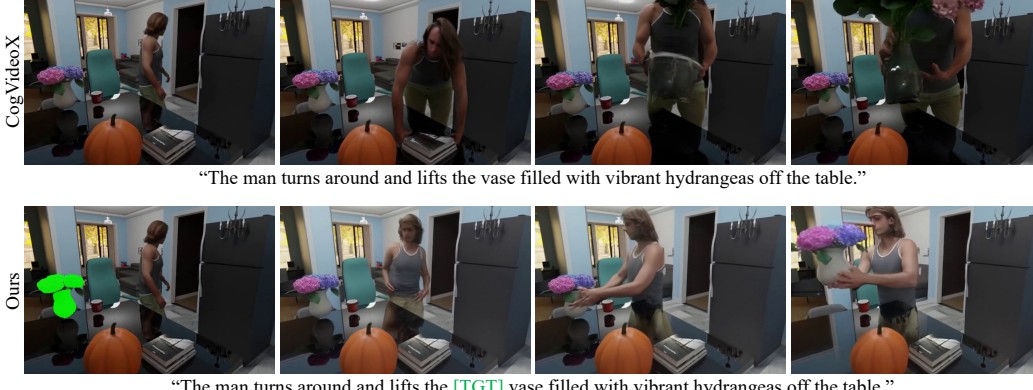

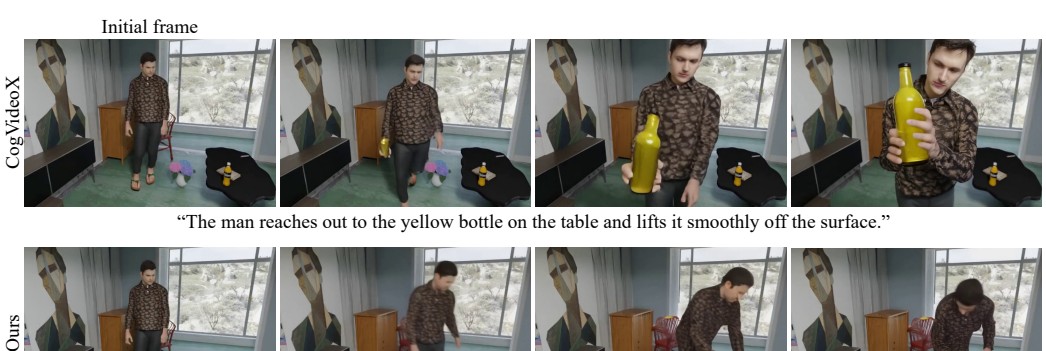

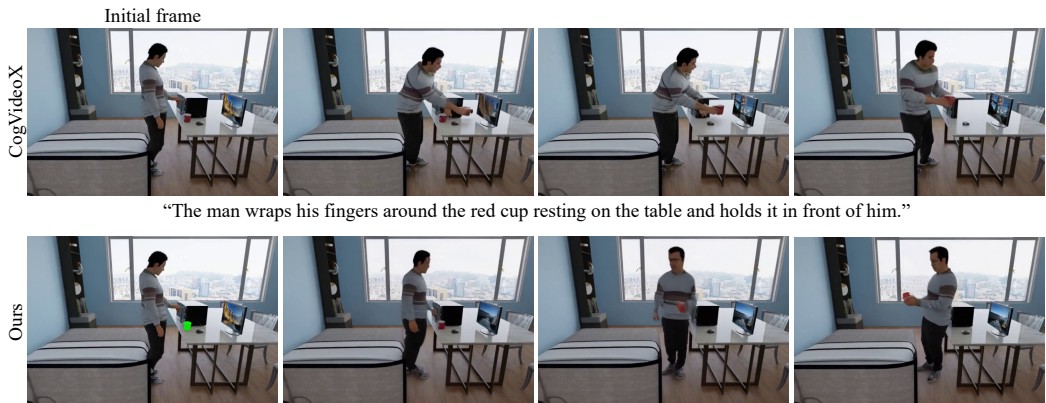

Figure 30: **Additional qualitative comparison on target alignment.** We compare results of original CogVideoX (Yang et al., 2025b) and our target-aware model. Our model successfully generates videos where the actor interacts accurately with the desired target. The target is colored in green every second row.

Initial frame

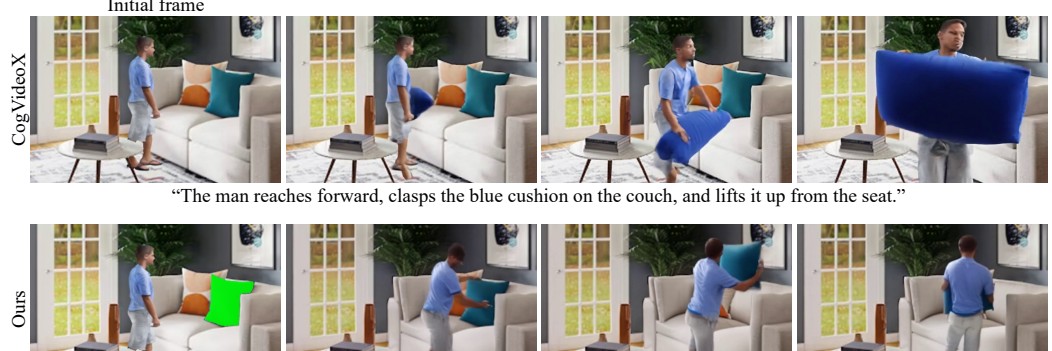

"The man reaches forward, clasps the blue cushion on the couch, and lifts it up from the seat."

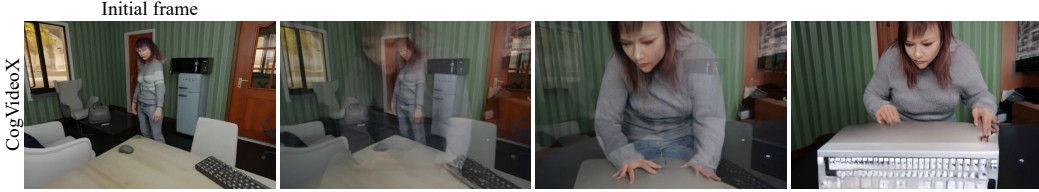

"The man reaches forward, clasps the [TGT] blue cushion on the couch, and lifts it up from the seat."

Initial frame

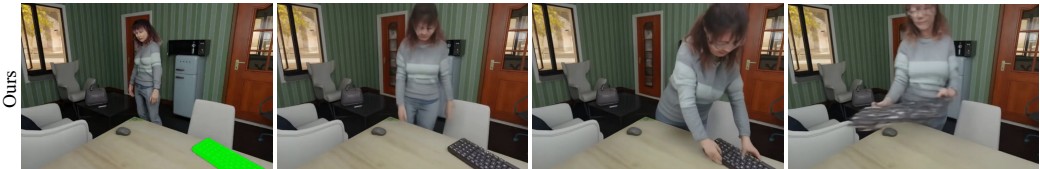

"The woman shifts her attention to the keyboard lying on the table and carefully lifts it up with both hands."

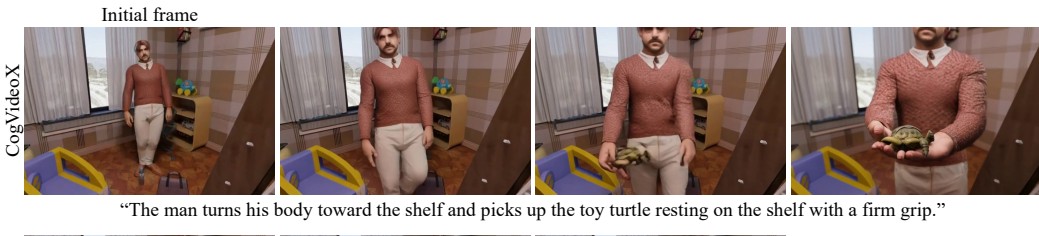

"The woman shifts her attention to the [TGT] keyboard lying on the table and carefully lifts it up with both hands."

Initial frame

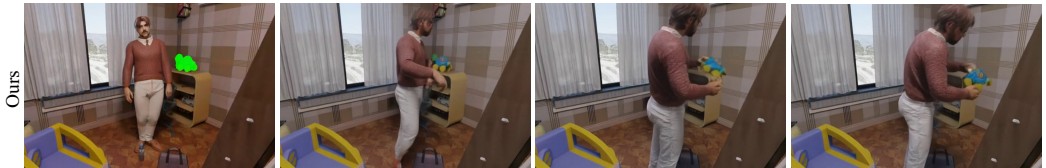

"The man turns his body toward the shelf and picks up the toy turtle resting on the shelf with a firm grip."

"The man turns his body toward the shelf and picks up the [TGT] toy turtle resting on the shelf with a firm grip."

Figure 31: **Additional qualitative comparison on target alignment.** We compare results of original CogVideoX (Yang et al., 2025b) and our target-aware model. Our model successfully generates videos where the actor interacts accurately with the desired target. The target is colored in green every second row.

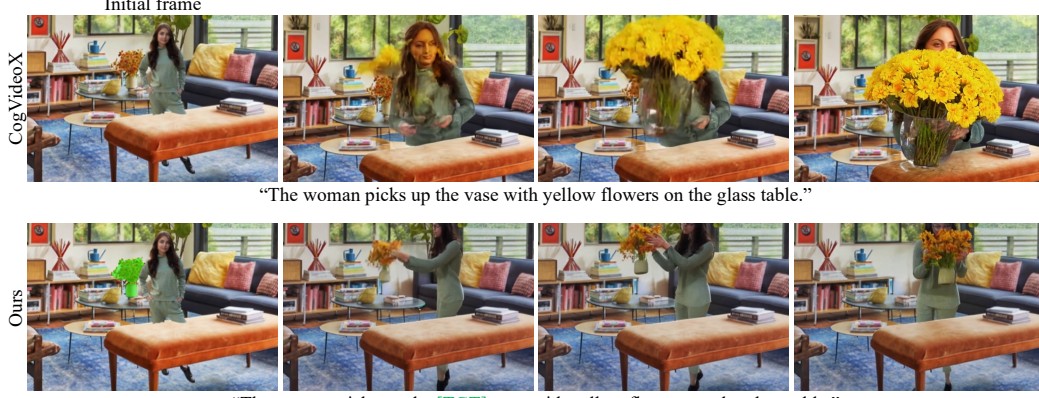

"The woman picks up the vase with yellow flowers on the glass table."

"The woman picks up the [TGT] vase with yellow flowers on the glass table."

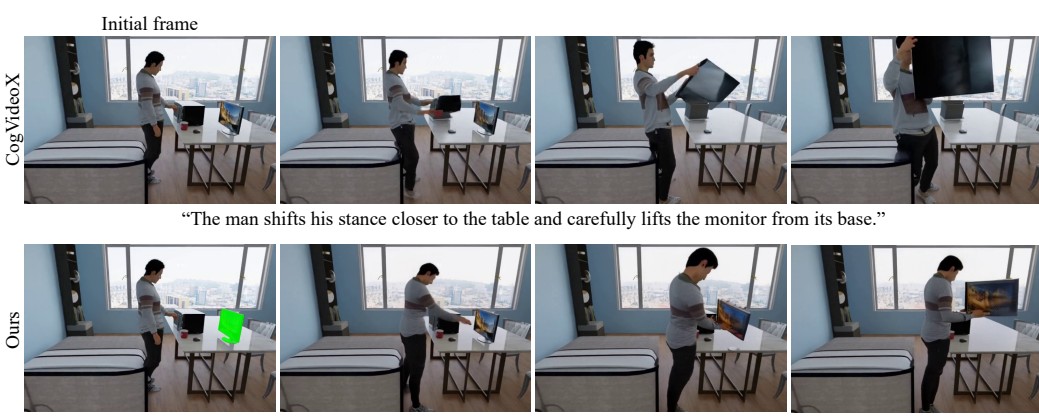

"The man shifts his stance closer to the table and carefully lifts the monitor from its base."

"The man shifts his stance closer to the table and carefully lifts the [TGT] monitor from its base."

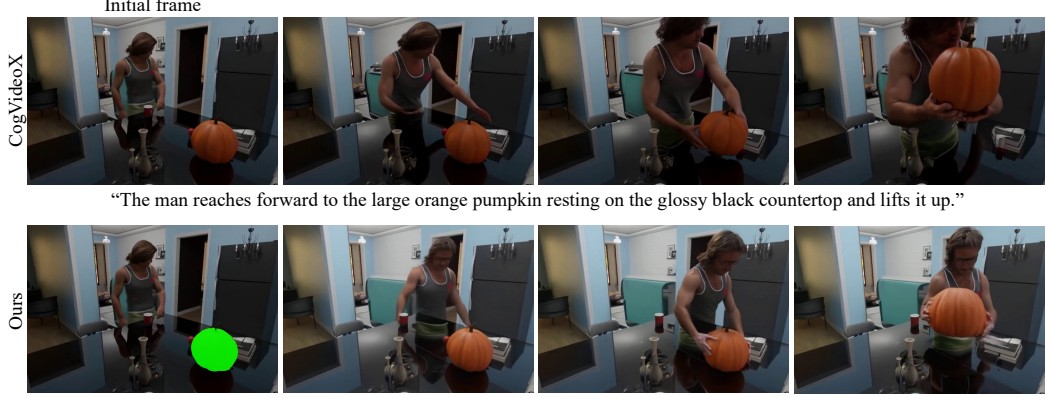

"The man reaches forward to the large orange pumpkin resting on the glossy black countertop and lifts it up."

"The man reaches forward to the [TGT] large orange pumpkin resting on the glossy black countertop and lifts it up."

Figure 32: **Additional qualitative comparison on target alignment.** We compare results of original CogVideoX (Yang et al., 2025b) and our target-aware model. Our model successfully generates videos where the actor interacts accurately with the desired target. The target is colored in green every second row.

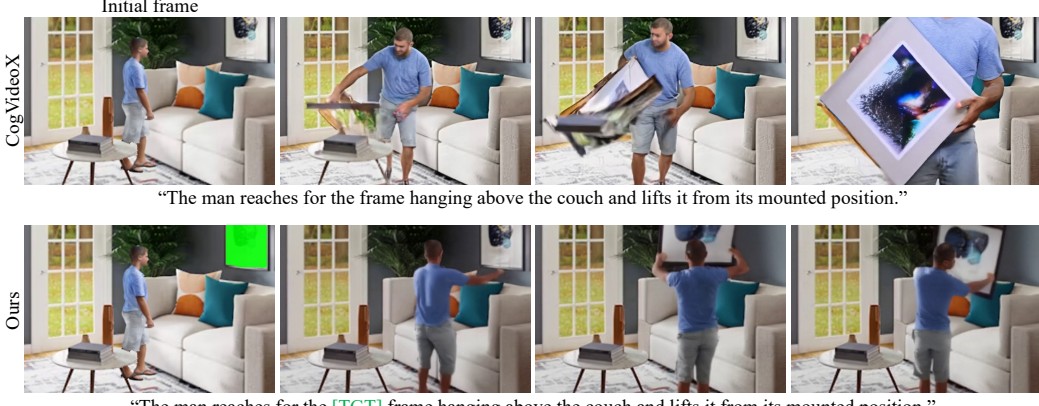

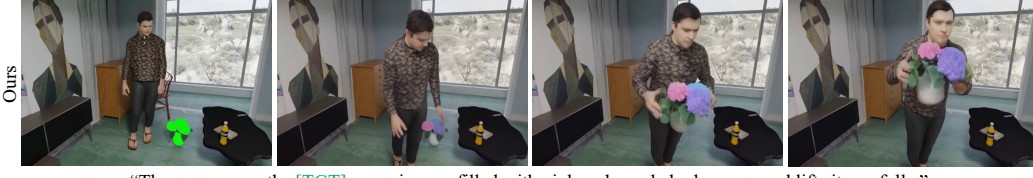

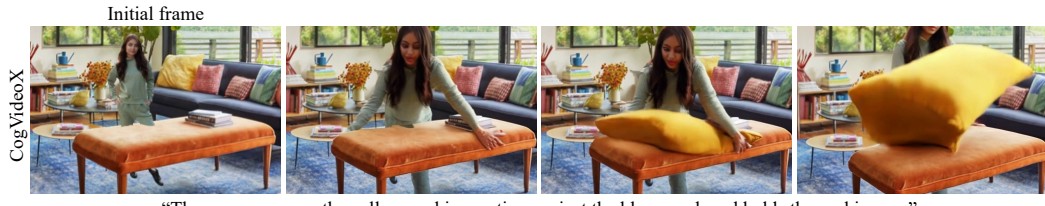

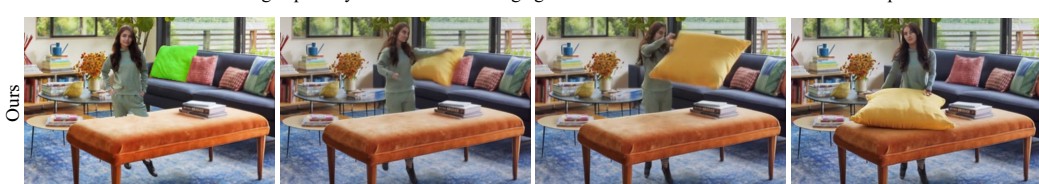

Figure 33: **Additional qualitative comparison on target alignment.** We compare results of original CogVideoX (Yang et al., 2025b) and our target-aware model. Our model successfully generates videos where the actor interacts accurately with the desired target.

