# OpenReview forum: "Target-Aware Video Diffusion Models"
_ICLR.cc/2026/Conference — ICLR 2026 Poster_

### Official Review · Reviewer_ceLm · 2025-10-31

**Soundness:** 3
**Presentation:** 3
**Contribution:** 3
**Rating:** 6
**Confidence:** 4

**Summary:**

This paper introduces a method to make image-to-video (I2V) diffusion models "target-aware", enabling them to generate videos where an actor plausibly interacts with a specific object present in the initial frame. The core problem identified is that existing I2V models, even when prompted, often fail to interact with the intended object and may hallucinate new ones. The proposed solution extends a base I2V model to accept a segmentation mask of the target object as an additional condition. The key contribution is a novel cross-attention loss applied during fine-tuning. This loss explicitly forces the model to align the cross-attention map of a special token with the spatial region of the input target mask. This alignment is selectively applied only to the video-to-text (V2T) cross-attention regions and to specific transformer blocks identified as most semantically relevant.
To train this model, the authors curate a new dataset from BEHAVE and Ego-Exo4D, consisting of 1290 video clips showing pre-interaction and interaction phases. Experiments demonstrate that this method significantly outperforms baselines in interaction accuracy (measured by a new "Contact Score" metric and extensive user studies) without degrading overall video quality. The paper also showcases two downstream applications: zero-shot 3D Human-Object Interaction (HOI) motion synthesis and long-term video content creation.

**Strengths:**

- The paper addresses a clear, practical, and important limitation of current video generation models. The "target-unawareness" of I2V models is a well-known failure case, and solving it is a key step toward using these models as effective "world models" or motion planners, as the authors suggest.
- The core idea of using a special token and enforcing its cross-attention map to match a target mask is shown to be effective. It directly injects spatial grounding into the text-conditioning mechanism of the transformer. The simplicity of the method (implemented as a LoRA module and an extra input channel) makes it practical and easy to adopt.
- Reasonable metrics are proposed and used, and decent improvements can be obtained.

**Weaknesses:**

- The curated dataset is quite small (1290 clips). While effective for this task, it may limit the diversity of learnable interactions. How well does the trained model generalize?
- The "Contact Score" is a good first-pass metric for this new task, but its definition is somewhat lenient. It is defined as a success if the contact detector finds an overlap with the target mask in at least one frame. This metric might not differentiate between a brief, accidental touch and a semantically correct, sustained interaction (e.g., a full "pick up and hold" action). This low bar could inflate the reported scores, though the qualitative results and user studies do suggest the generated interactions are indeed plausible.

**Questions:**

N/A

---

> ### Author Response · Authors · 2025-11-21
> **Response to reviewer ceLm (1/2)**
>
> **[W1] Dataset size for generalization**
>
> We appreciate the reviewer’s feedback regarding dataset size and generalization. We would like to clarify our core hypothesis regarding the modeling approach and dataset collection.
>
> Our work builds on the premise that large-scale pre-trained video diffusion backbones already have the expressive power to synthesize a wide range of plausible interactions (involving humans, animals, and objects). The missing component is not the ability to generate interactions, but the controllability to ground those interactions to a specific spatial target (i.e., to prevent the model from hallucinating new objects and forcing it to interact with the user-specified target).
>
> Accordingly, the purpose of our curated dataset is to align the spatial [TGT] token with the backbone's pre-existing generative priors to induce target awareness, rather than to expand the model's fundamental knowledge of interactions. In this context, a focused dataset of 1K+ clips is sufficient to learn this control mechanism. The visual diversity and physical realism are inherited from the pre-trained I2V backbone, not bounded by our fine-tuning data.
>
> Our empirical results strongly support this hypothesis. Despite being trained on a few human-centric indoor videos, our target-aware model successfully generalizes to scenarios completely unseen in our training set, demonstrating that it is leveraging the backbone's priors rather than overfitting to our data. We show qualitative examples in our original manuscript and also provide further examples (figures indicated in bold) in our revision, as summarized below:
> - Non-human interactions: Fig. 6, Fig. 8, **Fig. 15**, **Fig. 20**
> - Interactions in complex and cluttered scenes: Fig. 17
> - Interactions in scenes with largely different viewpoints: Fig. 18, **Fig. 21**
> - Interactions in outdoor scenes: **Fig. 16**
> - Diverse interactions with the same object: Fig. 19

---

> ### Author Response · Authors · 2025-11-21
> **Response to reviewer ceLm (2/2)**
>
> **[W2] Contact Score definition**
>
> We appreciate your careful analysis of Contact Score. We propose Contact Score as our primary metric because it directly reflects the core objective of our task: whether the model is aware of the target or not. For target-aware generation, even a brief contact with the right target is more important than a long, semantically rich interaction with a hallucinated object. To mitigate the reviewer’s concern, we have added two complementary metrics.
>
> First, we introduce *Contact Score $n$f*, a straightforward extension that considers an interaction successful only when the detected contact region overlaps the target mask for at least $n$ consecutive frames, explicitly discounting accidental or spurious touches. We note that this stricter criterion can sometimes deflate scores in cases where the interaction moves the target outside its initial mask region within a few frames, in which the contact naturally leaves the original target mask region even though the interaction is correct.
>
> Second, we introduce *Interaction Score*. We track points in the target-mask region of the first frame during video using CoTracker [1] and consider an interaction accurate only if both (1) the original Contact Score condition holds (the contact region overlaps the target mask in at least one frame) and (2) the mean displacement of the tracked points exceeds a certain threshold (10 pixels in our case, whereas static videos have mean displacement below 0.5 pixels). This jointly requires that the actor makes contact with the correct target and that the target undergoes nontrivial motion. It is important to retain the original Contact Score, because the displacement-based criteria alone can be triggered without correct interaction (e.g., the target moving without being touched). A limitation of this metric is that it can underestimate interactions where the target object remains largely static (e.g., pushing on a fixed object or sliding a hand over a surface), since the displacement term will remain small even when the interaction is correct.
>
> Nevertheless, the following table demonstrates that our method outperforms all baselines for all metrics.
>
> | Method            | Contact Score ↑ | Contact Score (2f) ↑ | Contact Score (3f) ↑ | Interaction Score ↑ |
> |:------------------|:---------------:|:---------------------:|:---------------------:|:--------------------:|
> | CogVideoX         |      0.560      |         0.446         |         0.377         |        0.474         |
> | CogVideoX w. data |      0.638      |         0.504         |         0.393         |        0.540         |
> | Attn. Mod.        |      0.546      |         0.399         |         0.340         |        0.455         |
> | Ours              |    **0.878**    |       **0.770**       |       **0.693**       |      **0.783**       |
>
> If the reviewer considers this extension appropriate, we are happy to incorporate these additional quantitative results, along with a discussion, into Appendix C of the revised manuscript. Moreover, as we refine our evaluation following your suggestion, we also find that the current Contact Score cannot be used to assess videos with non-human interactions, since the contact detector [2] only detects contacts involving human hands. We have added this limitation to Appendix C.4 (paragraph 6).
>
> [1] Karaev et al. "Cotracker: It is better to track together." ECCV 2024
> [2] Narasimhaswamy et al.  "Detecting hands and recognizing physical contact in the wild." NeurIPS 2020

---

### Official Review · Reviewer_AKUd · 2025-11-01

**Soundness:** 4
**Presentation:** 4
**Contribution:** 3
**Rating:** 6
**Confidence:** 5

**Summary:**

This paper addresses the problem of target-unaware video generation, where existing models fail to ensure an actor interacts with a specific target object designated by the user. The authors propose a method that feeds a target object's mask as an additional condition and introduces a novel cross-attention loss. During training, this loss forces the attention map of a special text token ([TGT]) to align with the input target mask. This allows the model to connect a text command to a specific spatial location in the scene, thereby generating target-aware interaction videos.

**Strengths:**

1. The paper is well-written, and the illustrative view clearly presents the entire pipeline.
2. The paper defines a clear and practical problem (target-aware video generation), It has excellent application value in multiple fields.
3. The ablation studies and quantitative analysis are thorough. The results clearly demonstrate the effectiveness of each module in the method and provides a reasonable explanation for the selection of hyperparameters.

**Weaknesses:**

1. Cross-attention loss cannot be considered as one of the innovations. There have already been numerous works in image generation and video generation that design losses at the attention level for fine-tuning models or optimizing generation results.
2. The amount of data used for training is insufficient. With only 1K+ video clips, the highly adaptable DiT architecture is prone to overfitting. Additionally, the data primarily consists of single-person indoor scenes, so the effectiveness of this method in outdoor or more complex scenarios requires further validation.
3. The examples demonstrating control over both the source actor and the target object are somewhat limited. It would be helpful to include more instances of interactions between objects (not just robotic arms) to showcase the generalization capability on the source actor.

**Questions:**

In the examples shown in the paper, the target object occupies a much smaller area relative to the actor. I am curious about how the model would perform when the target object takes up a significant portion of the frame (e.g., a car very close to the camera as the target object).

---

> ### Author Response · Authors · 2025-11-21
> **Response to reviewer AKUd (1/2)**
>
> **[W1] Cross-attention loss as innovation**
>
> We thank the reviewer for pointing this out and agree that cross-attention losses themselves are not new. We have revised the manuscript to avoid overstating this aspect, and we no longer describe our method as “proposing a novel cross-attention loss” or refer to it as “our cross-attention loss” in the sense of a new loss type. We mark our revisions in blue. Our contribution is instead to leverage a cross-attention loss to make the base video diffusion model target-aware. We ensure that the revised text reflects this more precise positioning of our contribution.
>
> ---
>
> **[W2] Dataset size for generalization**
>
> We appreciate the reviewer’s feedback regarding dataset size and generalization. We would like to clarify our core hypothesis regarding the modeling approach and dataset collection.
>
> Our work builds on the premise that large-scale pre-trained video diffusion backbones already have the expressive power to synthesize a wide range of plausible interactions (involving humans, animals, and objects). The missing component is not the ability to generate interactions, but the controllability to ground those interactions to a specific spatial target (i.e., to prevent the model from hallucinating new objects and forcing it to interact with the user-specified target).
>
> Accordingly, the purpose of our curated dataset is to align the spatial [TGT] token with the backbone's pre-existing generative priors to induce target awareness, rather than to expand the model's fundamental knowledge of interactions. In this context, a focused dataset of 1K+ clips is sufficient to learn this control mechanism. The visual diversity and physical realism are inherited from the pre-trained I2V backbone, not bounded by our fine-tuning data.
>
> Our empirical results strongly support this hypothesis. Despite being trained on a few human-centric indoor videos, our target-aware model successfully generalizes to scenarios completely unseen in our training set, demonstrating that it is leveraging the backbone's priors rather than overfitting to our data. We show qualitative examples in our original manuscript and also provide further examples (figures indicated in bold) in our revision, as summarized below:
> - Non-human interactions: Fig. 6, Fig. 8, **Fig. 15**, **Fig. 20**
> - **Interactions in complex and cluttered scenes**: Fig. 17
> - Interactions in scenes with largely different viewpoints: Fig. 18, **Fig. 21**
> - **Interactions in outdoor scenes**: **Fig. 16**
> - Diverse interactions with the same object: Fig. 19
>
> At the same time, we acknowledge current limitations of our method on complex scenarios in Appendix C.4. Since our current dataset contains masks covering only a single target, the model struggles when a single mask covers multiple objects. Our model also finds difficult to generate complex interactions involving multiple objects over time. Additional limitations and qualitative analysis are provided in Appendix C.4.

---

> ### Author Response · Authors · 2025-11-21
> **Response to reviewer AKUd (2/2)**
>
> **[W3] Limited examples of controlling both actor and target**
>
> We appreciate reviewer's suggestion to further demonstrate control over both the source actor and the target object. In addition to the robotic arms example in the original submission, we have provided additional qualitative results in Fig. 20 of our revised manuscript that showcase interactions between objects while simultaneously controlling the source actor and target object. In these examples, the source actor identity and appearance differ substantially from those seen during training, yet the model still produces plausible interactions with the specified target, as also illustrated in Fig. 15. These results better highlight the generalization capability on actors of our framework.
>
> ---
>
> **[Q1] Large targets in the frame**
>
> We thank the reviewer for raising this concern. To address this question, we have added qualitative examples in Fig. 21 where the target object occupies a large portion of the frame (e.g., objects close to the camera). In these cases, our model continues to interpret the mask with the [TGT] specification correctly and generates interactions focused on the large target region. However, we observe that as the target takes up a significant portion of the frame, the diversity of generated motions is reduced. We believe this stems from a limitation of our current setup: the model is trained on data with static cameras and therefore tends to generate videos with fixed camera trajectories (Appendix C.4 2nd paragraph). This prevents the camera from repositioning (e.g., zooming out) to better capture the interaction, which we identify as an interesting direction for future work.

---

### Official Review · Reviewer_Ph7a · 2025-11-03

**Soundness:** 3
**Presentation:** 3
**Contribution:** 3
**Rating:** 6
**Confidence:** 3

**Summary:**

This paper introduces a target-aware video diffusion model that enables a subject in an input image to interact with a user-specified target object using only a segmentation mask and a text prompt. The method extends a state-of-the-art I2V diffusion transformer (CogVideoX) by incorporating a target mask and introducing a special [TGT] token whose cross-attention maps are constrained to align with the mask via a cross-attention supervision loss. Selective application of this loss to specific cross-attention regions (V2T) and transformer blocks leads to stronger target conditioning. The paper also curates a dataset for this task and demonstrates applications such as zero-shot 3D HOI motion synthesis and long-term video creation. Experiments show clear improvements over baselines in target-interaction accuracy without degrading video quality.

**Strengths:**

+ Target-aware video generation for HOI and robotics planning is an important and timely direction. The paper identifies a real limitation in current I2V models (target ambiguity) and formulates a useful new capability.
+ Experiments systematically measure both target alignment and video quality, with diverse baselines and ablations. The Contact Score and user studies convincingly support the claims. Qualitative results are compelling, including multi-target, same-category objects, and generalization to animals and robotics scenes.

**Weaknesses:**

- The curated training dataset (1,290 clips) is relatively small and largely human-object centric. While results are strong, scaling considerations are not fully explored. It is unclear how well the method would generalize to more complex scenes or multiple simultaneous targets beyond simple masks.
- The paper hints at automatic mask extraction, but practical workflows for users remain unclear. Additional discussion on user burden or future plan for text-only grounding integration would be beneficial.
- Captioning uncertainty may cause noisy supervision; more detail on how noisy captions affect training would help strengthen the case.

**Questions:**

- How sensitive is the approach to inaccurate target prompts during training (e.g., captioning incorrectly naming the target)? Could self-refinement or contrastive prompts help?

- Could the proposed cross-attention loss be extended to temporal consistency (e.g., enforcing [TGT] attention along tracks across frames)?

- How does performance change if the target mask is inferred from a text-driven grounding model at test time instead of being provided manually?

---

> ### Author Response · Authors · 2025-11-21
> **Response to reviewer Ph7a (1/4)**
>
> **[W1] Generalization**
>
> We appreciate the reviewer’s feedback regarding dataset size and generalization. We would like to clarify our core hypothesis regarding the modeling approach and dataset collection.
>
> Our work builds on the premise that large-scale pre-trained video diffusion backbones already have the expressive power to synthesize a wide range of plausible interactions (involving humans, animals, and objects). The missing component is not the ability to generate interactions, but the controllability to ground those interactions to a specific spatial target (i.e., to prevent the model from hallucinating new objects and forcing it to interact with the user-specified target).
>
> Accordingly, the purpose of our curated dataset is to align the spatial [TGT] token with the backbone's pre-existing generative priors to induce target awareness, rather than to expand the model's fundamental knowledge of interactions. In this context, a focused dataset of 1K+ clips is sufficient to learn this control mechanism. The visual diversity and physical realism are inherited from the pre-trained I2V backbone, not bounded by our fine-tuning data.
>
> Our empirical results strongly support this hypothesis. Despite being trained on a few human-centric indoor videos, our target-aware model successfully generalizes to scenarios completely unseen in our training set, demonstrating that it is leveraging the backbone's priors rather than overfitting to our data. We show qualitative examples in our original manuscript and also provide further examples (figures indicated in bold) in our revision, as summarized below:
> - Non-human interactions: Fig. 6, Fig. 8, **Fig. 15**, **Fig. 20**
> - **Interactions in complex and cluttered scenes**: Fig. 17
> - Interactions in scenes with largely different viewpoints: Fig. 18, **Fig. 21**
> - Interactions in outdoor scenes: **Fig. 16**
> - Diverse interactions with the same object: Fig. 19
>
> Regarding the specific concern about multiple simultaneous targets under a single mask, we acknowledge this as a current limitation and appreciate the constructive feedback. As discussed in Appendix C.4, our current framework is optimized for single-target interactions. Handling multi-targets under a single mask or temporally shifting targets remains a future work. Accordingly, we have revised Appendix C.4 to include additional qualitative analysis (Fig. 24).

---

> ### Author Response · Authors · 2025-11-21
> **Response to reviewer Ph7a (2/4)**
>
> **[W2, Q3] Automatic mask extraction**
>
> We agree that practical mask acquisition is important. While fully automating the pipeline is not the main focus of our work, we implement and evaluate an automatic masking pipeline in Appendix C.1. Given an input target image and prompt, we use GPT-4o [1] to extract the noun and Grounded-SAM [2] to obtain the corresponding mask. This pipeline runs in \~15 seconds on a single RTX 3090 and achieves 93.14\% IoU on our evaluation set. When we feed these automatically generated masks into our model, the performance remains comparable to using manually prepared masks as shown in Tab. 7 (Contact Score 0.896 $\to$ 0.864, Video Quality 0.812 $\to$ 0.810). This shows that, in practice, users can rely on existing text-driven grounding tools with minimal manual effort. We further show that our model remains robust to quality or shape of the masks in Tab. 5 and 6.
>
> [1] Hurst et al. "Gpt-4o system card." arXiv preprint arXiv:2410.21276 (2024)
> [2] Ren et al. "Grounded sam: Assembling open-world models for diverse visual tasks." arXiv preprint arXiv:2401.14159 (2024)

---

> ### Author Response · Authors · 2025-11-21
> **Response to reviewer Ph7a (3/4)**
>
> **[W3, Q1] Noisy supervision due to captioning uncertainty**
>
> We appreciate the concern about caption noise and its effect on supervision. As mentioned in Section 4.4, our current training dataset includes noisy captions due to limitations of state-of-the-art video captioning tools [1], which often fail to correctly identify which object is actually being interacted with. Since we cannot manually annotate captions for all videos, we prepend a simple, but always true sentence, "The person interacts with [TGT] object." to each generated caption. This guarantees that the [TGT] token is semantically linked to the target under interaction, while the captioner-generated portion provides auxiliary information of appearance, scene context, and coarse motion of the video. These descriptive details help preserve the strong priors of the pre-trained backbone during fine-tuning.
>
> To assess the role of these noisy descriptive captions, we conducted an additional experiment where we fine-tune the model using only the general sentence, "The person interacts with [TGT] object.", completely removing the captioner-generated descriptions for all videos. As demonstrated in the table below, the variant trained only with the general sentence shows improvements in target awareness to some extent but exhibits degraded target alignment and video quality compared to the full setting with descriptions. This indicates that, despite their noise, the automatically generated captions still contribute for the model to be fully target-aware while maintaining its priors.
>
> | Method                                | Contact Score ↑ | Video Quality ↑ |
> |:--------------------------------------|:---------------:|:---------------:|
> | CogVideoX w. data                     |      0.638      |      0.810      |
> | Ours (general sentence only)          |      0.705      |      0.760      |
> | Ours (general sentence + description) |    **0.878**    |    **0.807**    |
>
> We further evaluate robustness to noisy captions during *inference* by corrupting the textual descriptions. In particular, we test (1) omitting the noun after [TGT], and (2) replacing the noun after [TGT] with an object name that does not correspond to the target and provide additional qualitative results in Fig. 22 of Appendix C.2. In both cases, we find that the model continues to generate interactions with the target specified with the mask, confirming that spatial grounding comes primarily from the mask-aligned [TGT] token rather than the exact noun. Together, these results suggest that our approach is robust to captioning uncertainty during both training and inference, and that further improvements (e.g., via caption self-refinement) can be incorporated as an orthogonal enhancement.
>
> [1] Yang et al.  "Cogvideox: Text-to-video diffusion models with an expert transformer." ICLR 2025

---

> ### Author Response · Authors · 2025-11-21
> **Response to reviewer Ph7a (4/4)**
>
> **[Q2] Temporal extension**
>
> Extending the cross-attention loss to enforce [TGT] attention along object tracks across frames is a very interesting direction. In this work, we intentionally constrain the loss to the first frame, so that the model learns a clear spatial binding with the target at the starting image while remaining free to generate diverse interactions with it over time, enabling new downstream applications such as the zero-shot robotic motion planning demonstrated in our paper. A natural next step would be to support temporal target switching: controlling which object is interacted with at different timesteps by providing different masks over time. Unfortunately, as discussed in Appendix C.4 (paragraph 5), our current model does not yet handle this case. When two different masks are provided for different timesteps, the model fails to switch to the second target, concentrating only on the first target. In the revision, we have added a qualitative analysis of target switching (Fig. 25) with [TGT] attention visualization along the temporal axis to highlight this limitation and to motivate temporal supervision as future work.

---

### Official Review · Reviewer_iau5 · 2025-11-04

**Soundness:** 3
**Presentation:** 3
**Contribution:** 2
**Rating:** 6
**Confidence:** 3

**Summary:**

This paper presents a target-aware video diffusion model that generates videos from an input image, mainly focusing on the human-object interaction generation. The key idea is to involve the target masks as new conditions with a special token. The experiments verify the effectivenss.

**Strengths:**

1. The idea is straightforward and reasonable.
2. The visualization is sufficient.

**Weaknesses:**

1. Integrating the mask condition is reasonable with expected improvement for interaction generation. Therefore, the technological innovation of this work is not significant. Using masks to control the video generation process is not novel. The author should reelaborate the insights of this paper.

2. The quantitative comparison is rather limited, making it less convincing. The author only compared one baseline. The influence of different backbones should be further analyzed to strengthen the claims of the paper.

3. I am confusing of the exact role of [tgt] token. In fact, [tgt] can simply encode the position information of the input mask as textual tokens. What are the benefits of this compared to not having the [tgt] token and only inputting the mask as the condition?

4. Providing the analysis of failure cases is beneficial.

**Questions:**

See Weakness.

---

> ### Author Response · Authors · 2025-11-21
> **Response to reviewer iau5 (1/3)**
>
> **[W1] Insights of the paper**
>
> We thank the reviewer for the chance to clarify the core insights of our work. We respectfully emphasize that our primary contribution is not the use of masks as an input modality, but the introduction of *target awareness* into video diffusion models.
>
> While we agree that masks are widely used in prior work [1, 2], the underlying objectives are fundamentally different. Previous approaches typically use masks (or layouts) to explicitly constrain subject trajectories or motion paths. In contrast, we use masks solely to indicate which object is the target of interaction, and rely on the video model to synthesize the necessary motion dynamics based on its own expressive priors. This distinction is critical: unlike previous approaches that require known actor motions or layouts as input, our framework generates these motions as outputs, enabling new downstream applications such as the zero-shot robotic motion planning demonstrated in our paper. The lack of existing datasets for this specific task further motivated us to curate a novel dataset to facilitate this line of research.
>
> Crucially, our experimental results demonstrate that simply integrating a mask condition is insufficient for target-aware interaction generation. As shown in Section 6.4, *Ablation Studies: Cross-Attention Loss Weight* (Tab. 4), when the cross-attention loss coefficient is set to $\lambda_{attn}=0$ (representing the model trained with the mask condition but without the cross-attention loss), the model fails to achieve target awareness. In fact, its performance is nearly identical to a standard CogVideoX fine-tuned without mask inputs (Tab. 1, second row). Significant improvements in target awareness only emerge when the cross-attention loss is active ($\lambda_{attn} > 0$), as also visually confirmed in Fig. 2(a). This confirms that our technical innovation lies not in the input format, but in the proposed mechanism that transforms the target-unaware model into a target-aware generator.
>
>
> [1] Yang et al. "Direct-a-video: Customized video generation with user-directed camera movement and object motion." SIGGRAPH 2024
> [2] Li et al. "DrivingDiffusion: layout-guided multi-view driving scenarios video generation with latent diffusion model." ECCV 2024

---

> ### Author Response · Authors · 2025-11-21
> **Response to reviewer iau5 (2/3)**
>
> **[W2] Limited quantitative comparison**
>
> We appreciate the reviewer’s concern about the breadth of quantitative comparisons and the influence of different backbones. We would first like to clarify that, within the CogVideoX backbone, the current submission evaluates several distinct baselines: (1) the original CogVideoX model, (2) CogVideoX fine-tuned on our dataset without the mask, and (3) an attention-modulation baseline adapted from prior work [1]. These baselines are designed to isolate different factors so that the effect of our proposed mechanism can be clearly assessed.
>
> Finding additional comparable methods for our specific setting is non-trivial. Most existing controllable video approaches either (1) focus on global layout or camera control [1, 2, 3], or (2) rely on heavy temporal conditioning such as 3D tracking or dense actor/target trajectories [4, 5]. These methods are optimized for different goals and assume much stronger supervision than a single target mask. In contrast, our method keeps the conditioning signal minimal and focuses on resolving which object the model should generate interaction with, rather than prescribing detailed motion paths.
>
> Regarding the influence of different backbones, our framework is conceptually backbone-agnostic: it only requires access to cross-attention maps and the ability to inject an additional mask channel, properties shared by many I2V diffusion models. In this work, we focus on a state-of-the-art open-source backbone (CogVideoX) and provide an in-depth analysis within this model. Specifically, we study which attention regions and layers to supervise (Tab. 2 and 3), how the cross-attention loss coefficient affects target awareness (Tab. 4), and how mask shape and quality impact performance (Tab. 5 and 6). We further test the model's robustness to automatically-generated masks (Tab. 7) and noisy captions (Tab. 8). We believe these ablations, together with the large margins over our baselines (Tab. 1), provide strong evidence that the proposed mechanism is effective and not overly sensitive to a particular configuration of CogVideoX. We acknowledge that extending the method to additional backbones would further strengthen our claims, and we view this as an important direction for future work.
>
> [1] Yang et al. "Direct-a-video: Customized video generation with user-directed camera movement and object motion." SIGGRAPH 2024
> [2] Li et al. "DrivingDiffusion: layout-guided multi-view driving scenarios video generation with latent diffusion model." ECCV 2024
> [3] He et al. "Cameractrl: Enabling camera control for text-to-video generation." ICLR 2025
> [4] Gu et al. "Diffusion as shader: 3d-aware video diffusion for versatile video generation control." SIGGRAPH 2025
> [5] Burgert et al. "Go-with-the-flow: Motion-controllable video diffusion models using real-time warped noise." CVPR 2025

---

> ### Author Response · Authors · 2025-11-21
> **Response to reviewer iau5 (3/3)**
>
> **[W3] Role of [TGT] token**
>
> We thank the reviewer for prompting us to clarify the role of the [TGT] token. While the input mask provides an effective spatial cue, it does not directly enter the text-conditioning pathway of the transformer, which makes it hard for the model to fully utilize the cue. The [TGT] token is introduced to bridge this gap. It serves as a text token that always refers to the region specified by the mask, as we align its attention with that region using the cross-attention loss. In this way, we directly inject spatial grounding into the text-conditioning mechanism, such that the model can explicitly use the target’s spatial information during generation. In the revised manuscript, we have added a clearer explanation of its role in L61-62 of the introduction.
>
> As mentioned in previous response to W1, our ablation in Tab. 4 shows that our proposed framework successfully brings target awareness to the model unlike the case of only inputting the mask as the condition.
>
> ---
> **[W4] Failure cases**
>
> Thanks for this constructive feedback! In Appendix C.4, we discuss limitations of our method, including cases where a single mask covers multiple objects and where the target specification changes over time. Following your suggestion, we have additionally included qualitative analysis of failure cases in the revised manuscript (Fig. 24, Fig. 25). We hope these examples provide further clarification of the model's boundary conditions and highlight current challenges for future work.

---

### Author Response · Authors · 2025-11-21
**General Response**

We would like to sincerely thank all reviewers (iau5, Ph7a, AKUd, ceLm) for their careful reading of our paper and for the constructive feedback, as well as the AC and SAC for coordinating the process. We truly appreciate the time and effort everyone has invested in evaluating our work.

We are highly encouraged to see the strengths of our submission highlighted by the reviewers:
- **Important and well-motivated problem**: "an important and timely direction .. formulates a useful new capability"(Ph7a), "clear and practical problem .. excellent application value in multiple fields"(AKUd), "addresses a clear, practical, and important limitation of current video generation models."(ceLm)
- **Simple yet effective core idea**: “idea is straightforward”(iau5), “core idea .. is shown to be effective, simplicity of the method makes it practical and easy to adopt”(ceLm)
- **Thorough analysis**: “diverse baselines and ablations”(Ph7a), “ablation studies and quantitative analysis are thorough”(AKUd), “reasonable metrics are proposed and used”(ceLm)
- **Strong results**: “qualitative results are compelling”(Ph7a), “decent improvements can be obtained”(ceLm)
- **Good presentation**: “visualization is sufficient”(iau5), “well-written, illustrative view clearly presents the entire pipeline”(AKUd)

with positive overall assessments: excellent(AKUd) and good(iau5, Ph7a, ceLm) soundness, excellent(AKUd) and good(iau5, Ph7a, ceLm) presentation, and good(Ph7a, AKUd, ceLm) and fair(iau5) contribution.


We are also highly grateful for the concerns and questions raised by the reviewers. These valuable suggestions help us improve the overall quality and clarity of the paper. In response, we have carefully revised and enhanced the manuscript as follows:
- avoid claiming a novel cross-attention loss and clearly state our contribution as leveraging cross-attention supervision for making the model target-aware
- clear explanation of the role of [TGT] token (Introduction)
- qualitative results on non-human interactions (Fig. 15, Fig. 20)
- qualitative results on outdoor interactions (Fig. 16)
- qualitative results on control over multiple entities (Fig. 20)
- qualitative results on large targets (Fig. 21)
- analysis on noisy captions (Appendix C.2, Tab. 8, Fig. 22)
- qualitative analysis of limitations (Fig. 24, Fig. 25)
- limitation of Contact Score (Appendix C.4 paragraph 6)

In the revised manuscript, newly added or modified content is temporarily highlighted in blue for ease of inspection during the discussion phase. We hope these updates address the reviewers’ concerns and help us clearly communicate the benefits of the proposed target-aware video generation framework to the ICLR community.

---

### Meta-Review · Area_Chair_NoiF · 2026-01-06

**Summary:**

* The main concern is limited novelty. Several reviewers note that conditioning generation on masks and using attention-level supervision are not entirely new ideas. In particular, the use of a cross-attention loss and mask-based control is seen by some as incremental unless more clearly articulated in terms of new insight or capability.

* A second recurring concern is the small and relatively narrow training dataset (≈1.3k clips), which is predominantly human-object and indoor focused. Reviewers question scalability, generalization to more complex or outdoor scenes, and robustness to diverse target configurations.

* Reviewers also asked for stronger empirical validation, including broader baselines, backbone analysis, and more failure cases.

* Reviewers also request for clearer justification of the [TGT] token’s role, more discussion on practical usage, and limitations of the Contact Score metric being relatively lenient.

**Reviewer Concerns:**

Concerns Likely Addressed in the Authors’ Response
* Clarification of the Role of the [TGT] Token. The authors explicitly justify the [TGT] token mechanism in the response.
* Metric Limitations (Contact Score Strictness). Results of two additional metrics are provided.

Concerns Partially Addressed:
* Stronger empirical validation. More failure cases are provided.
* Limited Dataset Size and Generalization. Authors argue that "a focused dataset of 1K+ clips is sufficient to learn this control mechanism". More examples showing generalization are provided. However, more data would be required to fully resolve this concern.

Concerns Still Likely Outstanding
* Novelty / Innovation Level. While the authors clarified the implementation and benefits, this conceptual concern remains and is not fully resolved.

**Reviewer Scores:**

Reviewer iau5

Original rating: 6 (marginal accept).

Likely change: increase to 8 or remain 6.

Reason: Clarification of the [TGT] token’s role, clarification of quantitative comparison, and better framing of insights partially address the concerns. However, the concerns about the limited quantitative comparison and insights of the paper are not fully solved.

Reviewer Ph7a

Original rating: 6 (marginal accept)

Likely change: increase to 8

Reason: Responses on noisy captions and mask workflows would address the corresponding concerns, while the concern on dataset size is partially addressed. The likely outcome is a slight positive update.

Reviewer AKUd

Original rating: 6 (marginal accept)

Likely change: remain at 6

Reason: Main concerns (novelty of attention loss, dataset size, generalization) are structural and not fully resolvable via rebuttal.

Reviewer ceLm

Original rating: 6 (marginal accept)

Likely change: 8 or 6

Reason: The metrics justification and additional metrics results would likely alleviate remaining doubts about evaluation leniency. The datasets size concern is partially addressed.

---

### Decision · Program_Chairs · 2026-01-26

Accept (Poster)